# Communicating climate change in a "post-factual" society: Lessons learned from the Pole to Paris campaign

Erlend M. Knudsen[1,2,a] and Oria J. de Bolsée[2]

[1]Institute for Geophysics and Meteorology, University of Cologne, Albertus-Magnus-Platz, 50923 Köln, Germany
[2]Pole to Paris, Christchurch, New Zealand
[a]now at: StormGeo, Nordre Nøstekaien 1, 5011 Bergen, Norway

*Correspondence to*: Erlend M. Knudsen (eknudsen@uni-koeln.de)

**Abstract.** The politicization of and societal debate on climate change science have increased over the last decades. Here, the authors argue that the role of climate scientists in our society needs to adapt in accordance with this development. We share our experiences from the awareness campaign Pole to Paris, which engaged non-academic audiences on climate change issues on the roads from the polar regions to Paris and through conventional and social media. By running and cycling across a third of the globe, the scientists behind the initiative established connections on the audiences' terms. Propitiously for other outreach efforts, the exertions were not in themselves the most attractive; among our social media followers, the messages of climate change science and action were more favourable, as measured by video statistics and a follower survey. Communicating climate action in itself challenges our positions as scientists, and here we discuss the impact such messages have on our credibility as researchers. Based on these reflections, as well as those from other science communication initiatives, we suggest a way forward for climate scientists in the post-factual society, who should be better trained in interaction with non-academic audiences and pseudoscepticism.

## 1 Background

The role of climate science in the public sphere has changed significantly since the mid-1980s. Following the formation of the Intergovernmental Panel on Climate Change (IPCC) and the U.S. Senate testimony of James Hansen in 1988, climate science has increasingly become a topic of political debate, media coverage and part of the daily discourse in our societies (Bolin, 2007; Ungar, 2016). Simultaneously, the scientific understanding of climate change has been rapidly expanding, with the number of climate change papers published per year growing exponentially (McSweeney, 2015) and the confidence in humans as the main cause of global warming has gone from insufficient to "extremely likely" (as defined by the IPCC First to Fifth Assessment Reports; Houghton et al., 1990; Stocker et al., 2013).

A corresponding increase has neither been seen in climate change legislation (Townshend et al., 2013), media coverage of climate change topics (Boykoff et al., 2018), nor in public perception of climate change (Capstick, et al., 2015; Zhao et al., 2016; Saad, 2017). Instead, the politicization and polarization of climate change has been growing, with the former referring

to how the science behind political decisions are increasingly promoted and attacked by advocates and opponents and the latter referring to the growing division between elites, organisations and political parties viewing climate change as a negative consequence of industrial capitalism and those opposing such views (McCright and Dunlap, 2011). This trend is arguably most notable in the U.S. (Capstick et al., 2015; Carmichael et al., 2017), where the partisan divide on environmental voting score (as defined by the League of Conservation Voters) grew from about 25 in 1970 to about 85 in 2015 (Dunlap et al., 2016). Since then, Donald Trump was elected as the country's 45th president and has repeatedly been questioning climate science, actively working against environmental legislation and funding of his predecessor and generally making the work of climate scientists more challenging (De Pryck and Gemenne, 2017; Alderman and Inwood, 2018; and references therein; Paasche and Åkesson, 2019, and references therein). A post-factual society has arisen, in which part of its members rather accept an argument based on their emotions and beliefs than one based on scientific facts (Leshner, 2007; Alvermann, 2017).

A post-factual political scene is not isolated to the U.S. alone; Brexit in the U.K. and the (re-)elections of Rodrigo Duterte in the Philippines, Andrzej Duda in Poland, Viktor Orbán in Hungary, Recep Tayyip Erdogan in Turkey and Jair Bolsonaro in Brazil are all examples of populistic solutions trumping science-based ones (Postel-Vinay, 2017; Paasche and Åkesson, 2019). Furthermore, the rise of social media has meant that everyone can act as journalists and editors in choosing what to post, where algorithms make sure to share posts from those with similar opinions, thus creating filter bubbles (Pariser, 2011; Alvermann, 2017; Bail, 2018). Conventional media can also reinforce filter bubbles by presenting scientific news within pre-existing worldviews of their audiences (Theel et al., 2013; Carmichael et al., 2017). Similar bubbles exist within academia, where scientists are trained to write for an already highly educated and specialized audience (Stiller-Reeve et al., 2016). Scientists are thus often seen as an elite without touch to the rest of society (Townson, 2016). For this reason, it is, more than ever, crucial to establish dialogues with those outside of academia in order to help trigger positive global changes (Leshner, 2007; Barnosky et al., 2016). Doing so, we, as scientists, need to choose our role within society carefully in consideration of the consequences for us individually and as a community (Pielke Jr., 2007; Vraga et al., 2018).

In this manuscript, we argue that the scientific community was not prepared for the intense politicization of climate change science (as defined by Zürn, 2014) that has occurred over the past several decades. However, we also contend that while climate polarization has reached new levels in the last few years (Dunlap et al., 2016), it is not too late for scientists to adapt to the highly charged political environment in which the very science of climate change is often discussed. Rapley and De Meyer (2014) argue that there is a gap between the role of the climate science community and the needs of society. As young environmental scientists having actively tried to bridge this gap, we share our experiences from climate change awareness initiatives, discuss their pros and cons, and discuss possible ways forward for the climate science community in terms of its interaction with society at large.

## 2 Our initiative: Pole to Paris

In early 2015, the authors co-established the non-governmental organization Pole to Paris. The aim of the initiative was to raise awareness of the threats posed by climate change, to people on our path as well as those reached virtually. However, it separated itself from most climate outreach actions by attempting to highlight the human-induced consequences of climate change rather than focusing on the pure scientific facts that underpin the reality of Earth's dynamic climate system. Following the unexpected collapse of the 15th Conference of the Parties (COP 15) to the United Nations Framework Convention on Climate Change (UNFCCC) in Copenhagen in 2009, the 2015 21st COP (COP 21) in Paris was regarded by many in the scientific, political and civil society communities as the last opportunity to begin to tackle climate change as a global community (Bäckstrand and Lövbrand, 2016; de Moor, 2017). Thus, the Pole to Paris project was purposefully timed leading to COP 21 in an attempt to galvanise support for a new global agreement in our wide society, as public awareness of climate change in a country is positively related to the unconditional climate mitigation targets of that country, as later suggested by Drummond et al. (2018).

The Pole to Paris project focused on reshaping the way scientists engage with the public on climate change issues. The nature of the problem – being a long-term process on a planetary scale – makes it difficult for individuals to grasp and engage with. In an attempt to remove this abstractness, we, as scientists, decided to hit the road in order to share climate science knowledge with people on the ground as well as collect their stories of experienced changes to share through our platforms. This allowed us to target audiences along the way not normally reached by scientific messages, meeting them face-to-face. Instead of inviting them to our universities, using a scientific jargon and sharing scientific information behind paywalls, we met them on their terms – in their home forums, using a familiar language and connecting through accessible formats.

To reach this audience, two journeys from the poles were mapped out: the 10,000-km long bicycle ride – the Southern Cycle – from Christchurch (New Zealand) and the 3,000-km long run – the Northern Run – from Tromsø (Norway), both finishing in Paris during COP 21 (Fig. 1). These journeys were led by two climate scientists, who left Christchurch and Tromsø shortly after completing their PhDs in Antarctic and Arctic climate change, respectively. 7.5 and 4 months later, respectively, they reached Paris, carrying flags from the melting polar regions and stories from people met along the way. The two were supported by eight other Pole to Paris team members, whose backgrounds ranged from environmental and political science to web and product design. While all members voluntarily and actively contributed to Pole to Paris by various means from their locations around the world, five of them also joined the main cyclist and runner for part of the journeys. Of the ten team members, only the main cyclist and runner were working full-time on the project (i.e., without getting paid), while the others had studies or jobs to balance simultaneously. We were all in our 20s, with the four female and six male team members representing eight different countries.

The public were invited to join the Southern Cycle and Northern Run journeys and actively engage in the climate dialogue in real time. This was partly done online through social media, partly at the events through open accessibility and partly on the roads themselves through planned and improvised meetings. To some extent, the latter happened because of GPS tracking on our website (Fig. 2), which allowed for other cyclists and runners to join us for part of the distances, providing an accessible and informal platform for face-to-face dialogues. The adventure component also helped to attract media attention, giving the project a platform to communicate the facts about climate change and the importance of COP 21 to the wider audience by engaging them in the journeys. Crucially, along the way, we held talks in schools, universities and many other public venues. To make our climate messages engaging, we called the teacher and read the local news ahead of the presentations to identify topics our audiences could relate to. The former also allowed for the students to be prepared for our presentations, following us online and learning about relevant material prior to our visit.

The ironic beauty of the climate change problem is that is encompasses the whole society, from health and food to tourism, migration and the economic system. Hence, we could always bring our climate messages into a familiar context for our audiences and thus stimulate their feedback. This was also helped by often starting presentations asking the audience what they already knew about the topic in a humane and positive attitude that set everyone at ease. Similarly, we created games and activity-based interactions, especially for our youngest audiences, which brought the large-scale climate problem down to his or her scale. Even though this took time from our given time slots, we found this to better adopt the pace and detail level of our climate messages while also lowering the threshold for questions and comments from the audience. Altogether, this created a true dialogue, in which we openly engaged the public to hear their perspectives and concerns about climate change before respectively responding to them, as suggested by Leshner (2003).

Our approach thus differed from the information deficit model, as outlined by Bucchi (2008). In this model, the public is considered passive and ignorant. Its hostility to science can be counteracted by appropriate injection of science communication, which is provided by experts (i.e., scientists) through a linear, one-way process to non-experts (the public) (Bucchi, 2008). However, this top-down approach is no longer appropriate for our current society, where science communication is addressing a wider agenda (Bucchi, 2008). Instead, the need and right of the public to participate in the scientific discussion has led to dialogue and knowledge models through which the involvement of lay people have enhanced the competencies of scientists and specialists (Callon, 1999; Trench, 2006). We found the latter models to be highly rewarding, as we learned a lot from the dialogues ourselves in addition to being better understood as communicators of scientific information.

We collaborated with our partners to create events, and we shared stories from the road through conventional and social media (Fig. 2). This provided a unique opportunity to interact with members of society not usually reached by the scientific

discourse. In line with O'Neill and Nicholson-Cole (2009) and Stoknes (2015), we highlighted the opportunities and inspiration of acting on climate change now rather than later. For example, from an economical viewpoint, strong, early climate action considerably outweighs its costs (Stern, 2007). Similarly, from a job market perspective, more jobs are added in the energy industry within renewables than are lost in fossil fuels (Fankhaeser et al., 2008). We still communicated the dangers associated with ongoing and expected consequences of climate change but in terms of relevant and experienced changes rather than fear rising from their cognitive dissonance following Extended Parallel Processing Model theory (Witte, 1992). This theory suggests that such messaging promotes a protection motivation and thus a willingness to change in accordance with the message for the recipient, in contrast to a defensive motivation and thus a reluctance to change (e.g., denial).

A conservative estimation is that more than one million people in 45 countries were reached through conventional and social media, which included 252 media outlets (thereof 15 blog posts written by us; Knudsen and de Bolsée, 2019) and almost 500,000 and 250,000 reached per Facebook post and Twitter tweet, respectively. While it is probable that some of our followers on Facebook, Twitter and Instagram overlapped, the breadth of conventional media coverage meant that we were able to reach a wider span of the society. For example, our story was featured five times on CNN in English, Spanish and Arabic, while Norwegian Broadcasting Corporation aired us 14 times. None of these are likely to be seen by the average Thai, Chinese or Indonesian, but our appearance in the Thai news channel TNN24, the China News Service or the Indonesian Jawa Pos might. Similarly, where coverage in the English-language news sources The Guardian, HuffPost or The Daily Star plausibly caught the attention of those already aware of human-induced climate change, the more domestic-focused Le Parisien in French, la Repubblica in Italian or Correio Braziliense in Portuguese almost certainly brought climate change into new light among their readers. Additionally, we gave 80 presentations in five languages along the running route alone.

## 3 Direct successes

Looking into the numbers from social media in more detail, the authors in 2018 conducted a statistical analysis on the reach of the videos created by Pole to Paris and shared through Facebook. Data for this analysis was fetched through the export function that Facebook offers for administered pages. In addition to information about the date videos were published, links to them and their titles, this function provides information about unique and total views, organic and paid views, and views after 3 seconds, at least 30 seconds (or to their end if that came first) and at 95 % of the video length (including viewers that skipped to this point). We subjectively categorized the videos by topic and main country(ies). Of the 42 total videos, we focused the analysis on the 32 in the most active period from June to December 2015. Detailed data on these can be found in Knudsen and de Bolsée (2019).

The 32 analysed videos spanned from 20 seconds to 6 minutes in length and showcased the life on the road from the Poles to Paris (i.e., challenges and joys of the run and bike ride), the various impacts associated with climate change along the way (e.g., coral bleaching in Australia from raising $CO_2$ levels and temperature, air pollution in China from carbon-intensive coal use, and glacial melt in Antarctica, Norway, and the European Alps from shifting precipitation patterns and increasing summer temperatures), and on the importance of climate action at COP 21 and home. Figure 3 shows the key results of the statistical analysis.

Of the 226 346 total video views after three seconds, 56 130 (25 %) were still there after 30 seconds and 16 703 (7 %) at 95 % of the video length (Fig. 3a). Of these views, 89 % (after three seconds) to 97 % (at 95 % of the video length) were unique (not shown), meaning that almost all videos were watched once. Similarly, the organic viewers (as compared to the ones reached through ads) were more enduring, accounting for 74 % of the views at 95 % video length compared to 58 % after three seconds (Fig. 3a). Sorted by topic, the climate action videos were on average the most popular by far, making up 82-87 % of the watched videos at the three video lengths (Fig. 3b). In comparison, the videos on the effects of climate change became relatively less popular over the length of the videos, comprising 11 % after three seconds and 8 % at their 95 % length. This contrasts the videos on the journeys themselves, which correspondingly rose from 6 % to 11 % of watched videos at the respective times.

The three most popular videos were thus, unsurprisingly, videos that promoted action on climate change through hopeful messages. The by far most popular video (with more than 100,000 views and a reach of nearly 500,000) focused how young inhabitants of Southern Pacific islands feel the effects of climate change through ongoing rising sea levels and get together to fight against it. This positive message of a younger generation working for an act on climate was the common theme for these three videos, which also included a more simply produced video on the motivation for why the main runner and cyclist left their offices in climate research to engage with the society at large (with almost 40,000 views and a reach of nearly 150,000). Out of our social media followers (more than 6,200 on Facebook, 1,200 on Twitter and 650 on Instagram), most of the Facebook ones were in the age group 25-34. This is perhaps explained by the fact that we were ourselves a team of millennials. Possibly more interestingly, the second largest group of followers was made of Generation Z (people born in the mid-1990s to the mid-2000s), pointing to the added reach of social media compared to other science communication tools, as also pointed out by Bowman et al. (2015).

As environmental scientists, who had tried to engage the people around us on climate change and biodiversity loss prior to Pole to Paris, the authors find the popularity of the climate action videos encouraging. However, this also questions our objectivity as scientists. Through the videos, we advocated for personal and societal action on climate change, as we did in media and our presentations. Hence, we moved beyond our core scientific base and took on roles as the 'science communicator' and 'the honest broker of policy alternatives,' as defined by Rapley and De Meyer (2014). We found this

necessary due to the nature of the problem – often seen as something far away in space or time. By sharing stories of climate change our audience could connect to, we made the problem more visible and graspable – to something right here, right now. This established connection also raised a willingness to do something about the problem, which we advocated for through the reduction of personal greenhouse gas emissions, through the investment power of consumers and companies and through

bringing the problem into light among family, friends and colleagues. Had we only communicated the threat of climate change without making it relevant and suggesting ways the listener could address the problem, we would have created a maladaptive response (e.g., denial) among our audience, according to Witte (1992).

Considering the time span over which the analysed videos were posted, the later videos were generally more popular. This

points to the increasing reach of Pole to Paris as the awareness project gained traction with kilometres covered, events held along the way, and mentions in the media. Even when the project reduced its activity after COP 21, the influence was still there, as exemplified by reaches of more than 100,000 on the less frequent Facebook posts in early 2016.

Correspondingly, while not posting regularly anymore, the authors were still able to reach some of Pole to Paris' followers

via our still active social media channels with a survey in 2018. The survey was set up through the online survey platform SurveyMonkey and asked the anonymous respondents a range of questions These included whether respondents followed Pole to Paris online, whether they learned anything new as a result of Pole to Paris and whether they found Pole to Paris to be a source of inspiration (Knudsen and de Bolsée, 2019). Interestingly, one of the key findings was that respondents were fairly evenly split on what they considered to be the most interesting aspects of the project. Several of the 37 respondents

highlighted more than one aspect, with 14 answers favouring the actual journeys from the Poles to Paris, 16 the same for the physical challenge of running and biking, 18 the scientific message on climate change, and 17 the human face that Pole to Paris put on climate change through stories from the ground.

In line with the statistical analysis of the Facebook videos, the fact that the scientific message was seen more interesting than

the journeys themselves, indicates that a project like Pole to Paris can find success in disseminating scientific information to a wider audience. Among other key findings from the survey, 31 out of 37 respondents reported that Pole to Paris inspired them in some way. This is also a strong indicator that unconventional projects in the vein of Pole to Paris can find success in connecting with non-scientific audiences in positive ways. Moreover, more than half (20 out of 37) indicated that they learned something new through Pole to Paris, signalling the potential that scientists have in bridging the gap between

academia and the public on fundamental societal issues.

Interpreting these numbers, one should keep in mind that the survey respondents already were followers of the climate awareness project Pole to Paris and thus not necessarily representative of the average population (Zhang et al., 2018). The three-year lag of this feedback to the project compared to its most active period also introduce some uncertainty of

remembrance and probably explain why less than 1 % of our social media followers responded to the survey. This small respondent rate meant that the answers did not necessarily represent those of a typical follower. Moreover, the time passed since their publications limited the statistical analysis here to Facebook videos, as other social media data were no longer available. Even so, we believe the numbers presented in this manuscript offer valuable insight on the worthiness of time spent on Pole to Paris and can help the outreach community in learning from our efforts.

## 4 Indirect successes

As also mentioned by Barnosky et al. (2014), the direct success of an initiative like Pole to Paris is almost impossible to quantify. Indirectly, the Pole to Paris team members took great value from being able to share climate science with our audiences and listen to their experiences of climate change. Engaging in two-way interaction with a range of audiences –

10 from farmers to senators, from preschool children to retirees and from Norwegians to Bangladeshis – provided invaluable insight to our own research questions, as also highlighted by Nisbet (2018). Fortunate with these encounters, we faced questions and concerns often far from ours, which opened our eyes and ears and widened our perspectives. As reported by Nisbet (2018) and references therein, we improved our communication and listening skills and extended our professional and social network. Both academic and non-academic members of society, especially the younger ones, expressed their

enthusiasm regarding the project. Both shared how it inspired them to find the courage needed to make changes in their own lives. The Paris Agreement, of which Pole to Paris was one of numerous initiatives building public support for, was arguably a better outcome of COP21 than the climate science community could have hoped for and, as later suggested by Drummond et al. (2018), might have been influenced by that awareness raised among people.

Schmid-Petri (2017) has argued that those in the scientific community who actively attempt to communicate the seriousness of climate change to a wide audience often are met with attempts "to discredit their scientific credibility, or to criticize the studies that are used or their underlying methods and models." As communicators of the scientific consensus, we inevitably experienced these tactics from climate sceptics in online forums. Mostly, the criticisms were from individual citizens and directed at us personally. Out on the roads to Paris, however, fact-based messaging was highly welcomed. Meeting people

where they are, in their own communities, communicating with them in their own terms, constantly trying to adapt our language to our audience, undeniably contributed to this. We connected through dialogue. Considering the politicized division of the media themselves (e.g., Brüggemann and Engesser, 2017), this positive experience of direct engagement supports the suggestion by Gauchat et al. (2017) that science participation and outreach could rebuild the credibility among communities most critical of scientists. Moreover, fostering constructive public conversations about science and society can,

among others, improve decision-making, promote trust and credibility in scientific findings and strengthen democratic processes (Wooden, 2006; Nisbet, 2018), ultimately countering politicization and polarization of science and post-factual movements, respectively.

Consequently, we worked hard to keep our credibility as researchers (Nordhagen et al., 2014), not partnering with organizations or initiatives on either of the climate advocacy fringes and not favouring one political party over another. Based on the feedback received, this scientific background and endeavour to remain objective allowed us to partner with
organisations otherwise out of reach, like the United Nations Development Programme (UNDP) and the World Meteorological Organization. Following the definitions by Nordhagen et al. (2014), we experienced a boost in personal and public credibility, more than outweighing a loss in professional credibility from our publication record hiatus while on the road, thus enhancing our researcher credibility overall. By being open about the role we played in public, we strove to negotiate the tension between our professional and public credibilities discussed by Nordhagen et al. (2014), in which our
goal of stronger climate action on a governmental level was challenged to some degree by the common academic view that researchers should remain detached from public policies. However, as Kotcher et al. (2017) point out, this notion is not supported by empirical evidence. On the contrary, in line with their results, we experienced no direct harm to our public credibility or to that of the scientific community.

We saw our role as awareness-raisers, increasing the understanding of climate science within all societal groups. Spanning the cultural differences within these groups, we tailored the message to the audiences in line with the suggestions by Somerville and Hassol (2011). These included framing climate change as a human and not only an environmental issue, focusing on the now instead of the decades ahead, leading with what we know, using a language adapted to a public discourse, being passionate, and connecting the dots between climate change and the personal experiences of the audience
themselves. We engaged the audience by illustrating what positive role they could play in averting the climate crisis.

For establishing personal connections to climate change among our audiences, we found that sharing personal experiences of climate change from people we met along the way was especially successful. As scientists, we are used to speak in terms like 2°C, 450 ppm and 50 cm, but most people cannot relate to these numbers. Rather, they relate to stories of people like them
whose livelihoods are threatened by climate change. Consequently, we listened to stories like those of a Sami, who might not be able to pass the reindeer herding tradition on to her children due to the warming winters; of a Bangladeshi, who might become a climate refugee due to the rising sea; and of a Londoner, who might be protected from the worst consequences in the metropolis but chooses to write about global environmental issues and work with organisations to find solutions. We shared these stories and others from the road through conventional and social media and in presentations on the way to Paris,
at a press conference and at the conference centre there and in a documentary and a TEDx talk since. Based on the video analysis alone, it is difficult to say that these messages were most popular, partly because we did not feature them all in videos and partly because they were both more and less popular than the videos featuring the scientists at the heart of the effort. However, based on interaction with journalists and our audiences, we have strong reasons to believe that these personal stories strongly helped in making the climate science relatable.

The nature of the Pole to Paris campaign allowed us to build an audience, which did not necessarily have a high interest in science nor necessarily a belief in climate change. This was purposefully done through several means: being on the road and therefore also meeting people who would not otherwise go to a talk about science on climate change; meeting university and

school students of all grades and consequently discussing with students who often had barely heard of the science behind climate change; and finally, running and biking, which invited participants for the physical challenge who would remain for the following talk on climate change and reached by a message they were not initially seeking. This point is also suggested by the number of the social media survey respondents indicating that through Pole to Paris they learned something new and got inspired (20 and 31 out of 37, respectively), which indicate that almost half of our followers were already literate on

climate change issues but did not know what to do about it. Even though the knowledge and interest in science differ between sociodemographic groups, as suggested by Schäfer et al. (2018), we found that all our audiences had a similar interest in learning about practical actions and solutions they could put to action at a personal level.

The ten languages spoken by the highly international Pole to Paris group members helped in this way by allowing us to

personally engage with a wide range of people on the roads from the polar regions to Paris. Besides, these language skills helped spread our messages even further, as suggested by the 62 % followers on Facebook speaking English, 16 % Indonesian, 6 % Norwegian, 4 % French, 3 % Spanish and 2 % German. Similarly, as suggested by Wooden (2006), the collaboration with local partner institutions (e.g., Gateway Antarctica in New Zealand, the Bjerknes Centre for Climate Research in Norway, the UK Youth Climate Coalition in the UK and Climate Generation in USA) offered experience for

successful ways of science communication within each country. This collaboration also allowed us to organize what we called the Global Voices events with our partner UNDP. These were set up outside the routes of the Northern Run and Southern Cycle (Fig. 1), during which youth came together to learn about climate change and how they could act upon it.

The experiences from Pole to Paris were, however, not unique. Other initiatives have been launched over the last few years

to increase climate change awareness and train scientists in more effective science communication. We were some of the 1.07 million people who took part in the March for Science on April 22, 2017. The series of rallies and marches defending the vital role science plays in our everyday lives was a direct result of the opposing direction on science policy taken by the current administration in the White House compared with its predecessor. However, March for Science has also been criticized, as it runs the risk of creating a false picture of scientists being more driven by ideology than evidence (Nature

supports the March for Science, 2017).

Furthermore, the authors have been involved in other more or less politically charged outreach projects. For instance, Climate Communication Cologne is an effort launched at the University of Cologne whose main objective is to facilitate science communication to a wide non-academic audience. This takes place in various forms, such as workshops, stand-up

comedy or videos, and in various arenas, from schools and universities to pubs and online communities. Another example is *Will You Hear Us*, a documentary on the tradition of caged birds in Indonesia, which has become unsustainable due to the ever-increasing demand for wild songbirds and poses a huge threat on biodiversity. Both authors are currently also writing comic books on climate change adaptation and mitigation and on biodiversity loss for high-school and elementary school students, respectively.

Common for all these initiatives is the eagerness to communicate science in ways that engage the layperson. To help us – and the reader of this manuscript – to learn from our efforts, we ideally would have set up a more standardized feedback scheme for our audiences during the active period of Pole to Paris. The feedback we did receive – in personal conversations and in online commentary forums – were most likely anomalously positive and negative, respectively. We could surely also have benefitted from more planning and training before undertaking these journeys, but this might have compromised the journeys themselves. Being the only two full-time-engaged team members, the two climate scientists of Pole to Paris – the lead cyclist and runner – had just completed their PhDs before taking on the journeys, while the other eight in the team had full time commitments to studies or employers to balance, which did not provide much room for further planning. This, along with the widely varying time zones the team members were based in and frequent lack of internet accessibility out on the Southern Cycle and Northern Run, meant that team meetings were less regular than what would have been ideal for making sure we were all pulling in the same direction.

Passion united the team and was contagious amongst our various audiences, creating better dialogues in a positive feedback loop (Nisbet, 2018). We cycled and ran out with rough plans and adapted along the way as engagement created opportunities (e.g., the Global Voices events and United Nations program partnerships) or disasters imposed limitations (e.g., the Nepal earthquake and Paris terror attacks). Similarly, even though we had scientific and professional communication training to start with, we learned a lot by doing. Most importantly, by meeting our audiences in running shoes, on a bicycle or in other informal settings, we connected as humans, which is critical for effective science engagement (Nisbet, 2018). While we strongly acknowledge the need for publishing research papers to further develop scientific questions, we emphasize that the findings thereof are incomplete if not shared with the society at large.

## 5 An adapted scientist

Based on the experiences outlined above, we identified some key components for successful science communication with non-academic audiences:

- Relevance

  Make sure your message is relevant to your audience and engage with them in familiar setting, with a familiar format and through a familiar language.

- Listening

  Let the audience ask questions and describe their understanding in their own words.
- Positivity

  Smile and try to focus on the possibilities rather than doomsday scenarios.
- Perseverance

  Learn by doing and keep doing it; all experiences are valuable.
- Passion

  For communicating science, knowledge of the topic is essential, but passion is the key for the audience to absorb it.

In our current society, we argue that the role of the 'pure scientist' (as defined by Rapley and De Meyer, 2014) is outdated and the need for the 'science communicator' and 'the honest broker of policy alternatives' (as outlined by Pielke Jr., 2007) is rising. The advancement of science might be of little significance if it is ignored by governments as well as laypeople and not suitably utilised by an educated society. Publishing an academic paper is unfinished business. As Barnosky et al. (2014) argue, it is only the beginning if our aim is to help society solve problems. However, current training of becoming scientists

does not fulfil society's current need for clear science communication and policy engagement (Leshner, 2007; Paasche and Åkesson, 2019). Thus, similar to Figueres et al. (2017), we argue that more emphasis should be placed on communication and media, policymaker and pseudoscepticism interaction training, and less on the published record.

For scientists at the beginning of their academic career, we support the notion by Leshner (2007), Brownell et al. (2013),

Rauser et al. (2017) and Nisbet (2018) that engagement in outreach activities helps shape the research questions, giving more effective tools for narrowing the widening gap between academia and the rest of society, and eventually providing a more constructive input for policy formulation on climate change. As we see it, this will act to reduce politicization and polarization of climate change, while also depressing the breeding ground for post-factual movements. Within academia, outreach training gives us better tools in teaching, mentoring of younger students and taking part in scientific discussions, as

well as contributing to better written research proposals and journal publications (Stiller-Reeve et al., 2016, and references therein).

Whether we like it or not, climate science and scientists have become part of the daily political and media discourse. Now it is up to us to adapt and play our new role objectively while keeping our credibility (as discussed by Nordhagen et al., 2014).

According to Rapley and De Meyer (2014), this has the potential to remove climate science from the direct firing line to leave the authority, responsibility and accountability for decisions transparently with the policymakers and the public. When done carefully, we have the potential, regardless of the audience's political preferences, to provide trustworthy information to the climate change discourse (Leshner, 2003; MacInnis et al., 2015; Hamilton, 2016). To prepare us for such a "wicked"

problem (as defined by Lorenzoni et al., 2007), we argue that communication training with actors beyond academia is indispensable.

## Data availability

The data used in this study is available in Knudsen and de Bolsée (2019).

## 5 Author contributions

EMK led the design and writing of the manuscript and carried out the statistical analysis. OJdB helped with writing and designed the social media survey.

## Competing interests

The authors declare that they have no conflict of interest.

## 10 Acknowledgements

We would like to thank the whole Pole to Paris team and everyone we encountered along the way. We also wish to thank the two reviewers and editors for constructive suggestions that improved the manuscript. We gratefully acknowledge the funding by StormGeo and the German Research Foundation (Deutsche Forschungsgemeinschaft; DFG) for the Transregional Collaborative Research Center "ArctiC Amplification: Climate Relevant Atmospheric and SurfaCe Processes, and Feedback
15 Mechanisms (AC)3" (TRR 172, project no. 268020496), which allowed EMK to write about our experiences for the benefit of science communicators globally.

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

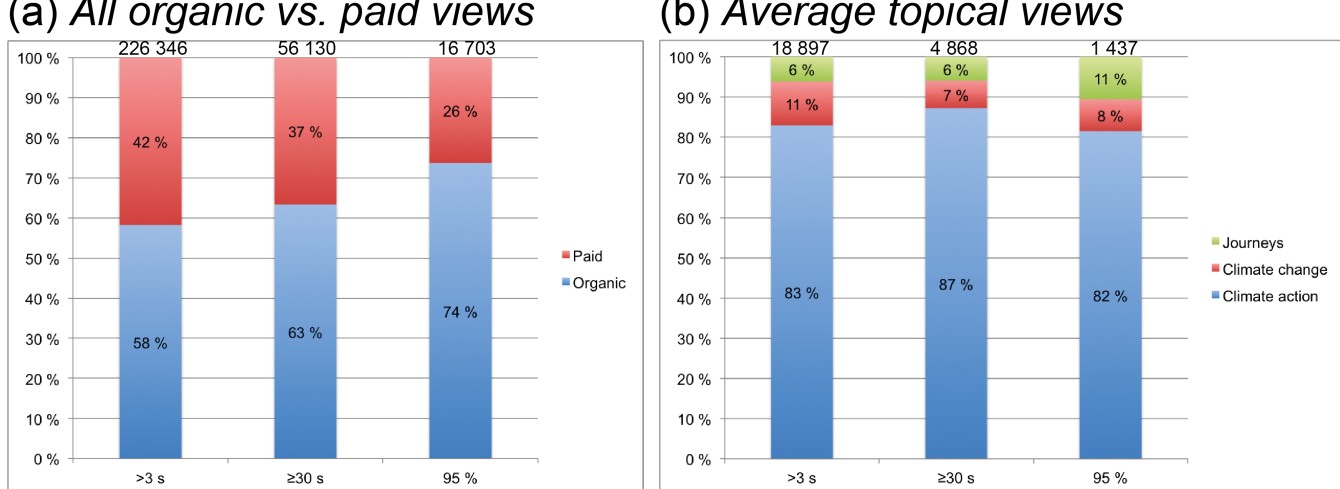

**Figure 2: Screenshots of Pole to Paris main website (upper left) and its web page about the Northern Run showing the interactive map with GPS tracking (upper right), Facebook (lower left), Twitter (lower centre) and Instagram (lower right) channels. Top row screenshots are from October 13, 2015, and bottom row from August 15, 2015, thus explaining the lower number of followers compared to the numbers presented in Sect. 3 from December 2015.**

**Figure 3: Percentages of total Facebook video views after three seconds (>3 s), at 30 seconds (or to the end, whichever came first; ≧30 s), and at 95 % of the video length (including people that skipped to this point; 95 %) for (a) organic (i.e., not paid; blue columns) and paid (red columns) views and (b) videos on climate action (blue columns), climate change (red columns) and the journeys themselves (green columns). Numbers above the columns in (a) and (b) represent total and average views, respectively.**