# Peer review of "Communicating climate change in a "post-factual" society: Lessons learned from the Pole to Paris campaign"

_Geoscience Communication, 2018_

## Referee Comment (RC1) · Salmon (Referee) · 29 Dec 2018

This is an interesting paper, telling an important story about the activities of (some number of?) young climate scientists in the lead up to the COP meeting in Paris. There are, however, some problematic aspects to the paper – I think it's trying to be more ambitious than it needs to be, and leans too heavily on data that's not representative of either the public, or the followers of the Pole to Paris journey that is the centrepiece of the manuscript. The second half of the paper was, to me, far more interesting – which focused more on the experience of the authors and situated this journey as one of a number of outreach initiatives about climate change that they're involved in. I would recommend reframing the paper in this context – as a case study of the experience of young climate scientists at this particular (and very important) time in history. In that

context, the bike ride, the march for science and the other outreach activities all tell a collective story – most importantly, that of the emerging climate scientists in this day and age (and possible changes to their training). The role of the scientist, in the field of climate change, has changed substantially over the last few decades and this narrative is a fantastic opportunity to showcase what a modern, engaged, climate scientist might look like. It draws on experiences and offers useful recommendations as well as being situated in the relevant literature (although there is an opportunity to draw on this more, I appreciate this in only a short paper).

I have provided some feedback directly on a PDF of this document that I am happy for the authors to see but it is quite specific – if my recommendations above are considered, then many of those comments will likely become redundant.

I wish the authors all the best with the publication of this manuscript, which portrays an important aspect of being a climate scientist that is rarely documented outside of social media, and even more rarely examined critically. Were there an opportunity to expand on the length of this paper (which I don't necessarily encourage), then I would also like to see greater reflection on how the initiatives could be improved, and how the authors would learn from their experiences, were a similar opportunity to arise.

In terms of the specific questions posed by the journal: 1. Does the paper address relevant scientific questions within the scope of GC? Yes

2. Does the paper present novel concepts, ideas, tools, or data? Yes

3. Are the scientific methods and assumptions valid and clearly outlined? Not entirely

4. Are the results sufficient to support the interpretations and conclusions? Not entirely – although a reframing of the argument would remedy this. The data presented is unfortunately not that compelling, which is often the case with outreach evaluation exercises (and not necessarily the fault of the authors).

5. Do the authors give proper credit to related work and clearly indicate their own new/original contribution? To an extent, yes

6. Does the title clearly reflect the contents of the paper? No – the current title, "The role of climate scientists in the post-factual society" implies a greater discussion and consideration of the role of the climate scientist, rather than a report on the outreach these particular scientists did. As suggested above, I am recommending a re-framing of the piece that would potentially mean the title could remain the same. I really like the title, and the aspiration of the paper to explore/ argue that the "scientific community was not prepared for the intense politicization of climate change" (para 2), but I don't think the paper in its current form lives up to these. Furthermore, considering "post-factual society" is a phrase used in the title, little time is given to defining or exploring this important concept.

7. Does the abstract provide a concise and complete summary? Broadly, yes. As above – I'd love to see the title and abstract stay the same, and the content be adapted to deliver on these.

8. Is the overall presentation well structured and clear? Adequate

9. Is the language fluent and precise? Some areas need tightening up or a proof-read, as identified in my comments on the PDF

10. Are the number and quality of references appropriate? The number is appropriate but the breadth could be expanded – the majority of these references come from physical/ climate science/communication journals. Considering the focus of this article, I would expect to see a few more references from the field of Public Engagement with Science, especially related to dialogue and two-way interaction, which is referred to several times.

Please also note the supplement to this comment:
https://www.geosci-commun-discuss.net/gc-2018-16/gc-2018-16-RC1-supplement.pdf

**Supplement:**

[revised manuscript text omitted]

---

## Short Comment (SC1) · 2 Jan 2019

"The role of climate scientists in the post-factual society," describes the authors' outreach efforts during the Poles to Paris campaign and their efforts to generate awareness about climate change leading up to the 21st COP in Paris. The authors' reflections on this experience and analysis of their social media posts are used to generate several recommendations for scientists on how outreach should be conducted and the role of scientists in society, particularly in the context of the polarized and politicized topic of climate change science. I commend the authors for undertaking such an ambitious outreach effort and for taking the time to assess and reflect on their efforts. I also commend the authors for using data - albeit limited - to further reflect on the effects of their efforts. Such commentaries are important for facilitating a broader discussion in the

scientific community about where the institution of science has been, and where it may be heading, and its ever-evolving role in society.

The most important finding in this paper, in my opinion, is that the climate action message - coming from climate scientists - was the most popular (among social media video viewers). This finding is not surprising, considering communication theories like the Extended Parallel Processing Model (Witte) suggest that increasing efficacy to cope with risk is a critical part of effective risk message processing. However, this finding (or reflection) is most interesting to me considering that many climate scientists are hesitant to venture past explaining the causes and effects of climate change, into discussing solutions, for fear of being seen as an advocate and losing credibility. The authors could strengthen this paper by spending more time reflecting on and discussing the content of their climate action messages and exploring where those messages sit on a spectrum from objective to advocate. Furthermore, these reflections could be more strongly placed in the existing literature about the role of scientists in society. The authors say on page 1, line 13, "The role of climate science in the public sphere has changed significantly since the mid-1980s." I would like to hear more about this line of reasoning, and I recommend the authors explore some of the existing literature from science and technology studies that reflects on the role of scientists in society (i.e. The Honest Broker, by Roger Peilke Jr., recent work by John Kotcher et al. also explores scientists' advocacy messages).

While I appreciate the presentation of the data related to the campaign, I encourage the authors to be much more transparent about who exactly was engaged in the different elements of the campaign and where they have data and where they do not. On page 2, line 23, the authors said, "A conservative estimation is that more than one million people in 45 countries were reached through conventional and social media." If it is included, I would like to see a much more detailed description of how this figure was produced. Is this based on social media impressions? Is this based on traditional media circulation rates? The analysis of the social media videos is interesting,

but the authors should acknowledge the extent to which this audience is similar to or distinct from the audiences who participated in public lectures and those who engaged with the campaign through traditional news sources and the population as a whole. To that end, I would strongly recommend the authors avoid the use of the term "general public." From the description provided, several distinct audiences were targeted and reached during the campaign (i.e. school children, people who attended a lecture, people who watched a video on Facebook) - and each of these audiences likely has unique characteristics that are relevant when considering the authors' final outreach recommendations. In particular, I encourage the authors to address the extent to which their campaign attracted people who already had a high interest in science or belief in climate change (see Besley, "Audiences for Science Communication" for further discussion from a US context).

This paper makes an important observation about the need for scientists to engage in dialogue, especially face to face communication. These recommendations are aligned with a recent report by Matt Nisbet, for AAAS (https://www.aaas.org/sites/default/files/s3fs-public/content_files/Scientists%2520in%2520Civic%2520Life_FINAL%2520INTERACTIVE%2520082718.pdf), which explains the need for scientists' engagement in civic life. Discussing this would also be valuable in the context of the role of scientists/science in society. However, I am not persuaded by the authors' assertion that their outreach efforts are the solution to climate change polarization, politicization, and the "post-truth" world. First, I think there needs to be stronger evidence of which audiences were reached in the campaign in order to make this claim. Secondly, I think these terms must be defined and explicated if they are to be used to generate recommendations for scientists. For example, what are the causes of politicization, and why do the authors think this particular outreach approach helped resolve it? Similarly, what are the causes of polarization (it is distinct from politicization), and do the authors think the campaign helped to overcome this? Why? How? Furthermore, due to the international nature of the campaign, it would be useful to understand how the effects of the outreach varied

between different nations—because politicization and polarization likely vary widely amongst the different nations included in the campaign.

In summary, I feel that this is an important commentary and that the authors made a very good effort to describe their outreach activities, who was reached and provide useful reflections on the lessons learned. To that end, I think this paper is perfectly suitable for Geoscience Communication. However, the paper can be strengthened by further reflection on the role of scientists in society, and how the effects of this campaign align and diverge from prior research related to this role. The outreach recommendations are good, but the readers will be better able to assess the validity of the recommendations to their situation if more information is provided about specifically what audiences were engaged in the campaign's messages and activities, why the authors targeted these audiences, and what observations were made about the success of the campaign (without over-relying on the limited data available). In my opinion, the weakest part of this paper is the link to the post-truth environment, polarization, and politicization. While the campaign may have had positive influences in these areas, these are highly complex social processes that probably varied across the nations involved in the campaign. A much more sophisticated analysis would need to be undertaken to understand the effects of the campaign on these social processes.

In terms of the specific questions posed by the journal:

1. Does the paper address relevant scientific questions within the scope of GC? Yes

2. Does the paper present novel concepts, ideas, tools, or data? Yes

3. Are the scientific methods and assumptions valid and clearly outlined? No. Please see prior comments related to clearly articulating who was among the campaign's target audiences and where is data to support the claims for each audience.

4. Are the results sufficient to support the interpretations and conclusions? No. However, I agree with the other reviewer – that shifting the focus to the role of scientists and

reflecting more on the experience, would remedy this.

5. Do the authors give proper credit to related work and clearly indicate their own paper new/original contribution? As explained, I think the authors could rely more on prior work in science and technology studies related to the role of science in society and how this has evolved over time.

6. Does the title clearly reflect the contents of the paper? No. As I explained, delving into a discussion about the "post-truth" era, polarization, and politicization requires much more explication and a different kind of data than what is provided here. I suggest renaming the paper.

7. Does the abstract provide a concise and complete summary? Generally, it does, but this paper could evolve based on the reviews and I anticipate the abstract will change accordingly.

8. Is the overall presentation well-structured and clear? As it stands, I think this paper tries to do too many things at once and that makes the structure somewhat hard to follow. I anticipate that when the focus is narrowed, the structure and flow will also improve.

9. Is the language fluent and precise? It is okay, but the descriptions of the audiences reached through the effort must be more precise. In general, I encourage the authors to explore the meanings of terms like polarization and politicization if they continue to use them.

10. Are the number and quality of references appropriate? Generally yes, but I think this paper would benefit from a deeper literature review on the role of science in society and some references from science and technology policy studies.

---

## Author Comment (AC1) · 7 Feb 2019

**The role of climate scientists in the post-factual society**

Erlend M. Knudsen and Oria J. de Bolsée

**Set-up of response**

We thank the reviewer for her suggestions on the manuscript. With the changes explained below, we feel that the paper is strengthened compared to its first submission.

In the following, we go through each comment by the reviewers (reproduced here in gray text for your reference) and explain our choices of changes in accordance with these. Where changes to the text in the manuscript are made, the relevant excerpt is reproduced from the .pdf manuscript to this .docx response in *italic text*, with changes written in *italic green text*.

**Response to the review by Rhian Salmon**

Reframing the manuscript

The second half of the paper was, to me, far more interesting – which focused more on the experience of the authors and situated this journey as one of a number of outreach initiatives about climate change that they're involved in. I would recommend reframing the paper in this context – as a case study of the experience of young climate scientists at this particular (and very important) time in history. In that context, the bike ride, the march for science and the other outreach activities all tell a collective story – most importantly, that of the emerging climate scientists in this day and age (and possible changes to their training).

We appreciate you sharing your opinions and recommendations on this matter, which we believe help us strengthen the manuscript. We have therefore tried to follow your suggestions as much as possible in the updated manuscript. Here, we tell the story of being young environmental scientists having tried to actively bridge what we see as a widening gap between science and populism, building on our experiences from Pole to Paris (the two authors ran about 2450 km and 750 km of the Northern Run in addition to backing other parts of the project, including the Southern Cycle) and other environmental awareness projects that we have been involved in.

In line with this, we have done several changes to the text, as explained below and highlighted in the updated manuscript. We have also changed the manuscript title to reflect this storyline, from "The role of climate scientists in the post-factual society" to "The role of climate scientists in the post-factual society: Reflections from the awareness campaign Pole to Paris". With these changes, we hope and believe that the manuscript now tells a clearer story, in which we share our experiences to contribute to the scientific discussion on what role climate scientists should consider playing in the 21st century.

Reflection on how to improve

Were there an opportunity to expand on the length of this paper (which I don't necessarily encourage), then I would also like to see greater reflection on how the initiatives could be improved, and how the authors would learn from their experiences, were a similar opportunity to arise.

Based on your suggestion, we have added two paragraphs to the end of Sect. 4 that read:

*Common for all these initiatives is the eagerness to communicate science in ways that engage the layperson. To help us – and the reader of this manuscript – learn from our efforts, we ideally would have set up a more standardized feedback scheme for our audiences during the active period of Pole to Paris. The feedback we did receive – in personal conversations and in online commentary fora – were most likely anomalously positive and negative, respectively. We could surely also have benefitted from more planning before undertaking these journeys, but this might have compromised the journeys themselves. Being the only two fully "working" (i.e., without getting paid) on the project, the two climate scientists of Pole to Paris – the lead cyclist and runner – had just completed their PhDs before taking on the journeys, while the other eight in the team had full time commitments to studies or employers to balance, which did not provide much room for further planning. This, along with the widely varying time zones the team members were based in and often lack of internet accessibility out on the Southern Cycle and Northern Run, meant that team meetings were less regular than what would have been ideal for making sure we were all pulling in the same direction.*

*Passion united the team and contaminated our various audiences, creating better dialogues in a positive feedback loop (Nisbet, 2018). We cycled and ran out with rough plans and adapted along the way as engagement created opportunities (e.g., the Global Voices events and United Nations program partnerships) or disasters imposed limitations (e.g., the Nepal earthquake and Paris terror attacks). Similarly, even though we had scientific and communicational training to start with, we learned a lot by doing. Most importantly, by meeting our audiences in running shoes, on a bicycle or over a beer, we connected as humans, which is critical for effective science engagement (Nisbet, 2018). While we strongly acknowledge the need for publishing research papers to further develop scientific questions, we emphasize that the findings thereof are incomplete if not shared with the society at large.*

Justification of title

I really like the title, and the aspiration of the paper to explore/ argue that the "scientific community was not prepared for the intense politicization of climate change" (para 2), but I don't think the paper in its current form lives up to these. Furthermore, considering "post-factual society" is a phrase used in the title, little time is given to defining or exploring this important concept.

Thank you for pointing out the mismatch between the title and the manuscript itself. In the updated manuscript, we have changed the title from "The role of climate scientists in the post-factual society" to "The role of climate scientists in the post-factual society: Reflections from the awareness campaign Pole to Paris". Furthermore, we have expanded the first paragraph in Sect. 1 into three to better explain what is meant with politicization and polarization of climate change and a post-factual society.

As a result, the first three paragraphs in Sect. 1 now read:

[revised manuscript text omitted]

Breadth of references
The number is appropriate but the breadth could be expanded – the majority of these references come from physical/ climate science/communication journals. Considering the focus of this article, I would expect to see a few more references from the field of Public Engagement with Science, especially related to dialogue and two-way interaction, which is referred to several times.

Based on your suggestion, we have added several new references to the manuscript with a special focus on the field of public engagement with science. While we do not provide a complete list of new references here (these are highlighted in the updated manuscript), we emphasize the most relevant ones.

Excerpt of the 2[nd] paragraph in Sect. 1:
*For this reason, it is, more than ever, crucial to establish dialogues with those outside of academia in order to help trigger positive global changes (Leshner, 2007; Barnosky et al., 2016).*

Excerpt of the 2[nd] paragraph in Sect. 2:
*Crucially, along the way, we held talks in schools, universities and many other public venues and were joined by other cyclists and runners for part of the distances. This created a two-way communication, in which we openly engaged the public to hear their perspectives and concerns about climate change before respectively responding to them, as suggested by Leshner (2003).*

Excerpt of the 1[st] paragraph in Sect. 4:
*Engaging in two-way interaction with a range of audiences – from farmers to senators, from preschool children to retirees and from Norwegians to Bangladeshis – provided invaluable insight to our own research questions, as also highlighted by Nisbet (2018). Fortunate with these encounters, we faced questions and concerns often far from ours, which opened our eyes and ears and widened our perspectives. As reported by Nisbet (2018) and references therein, we improved our communication and listening skills and extended our professional and social network.*

Excerpt of the 2[nd] paragraph in Sect. 4:
*Moreover, fostering constructive public conversations about science and society can, among others, improve decision-making, promote trust and credibility in scientific findings and strengthen democratic processes (Wooden, 2006; Nisbet, 2018), ultimately counteracting politicization and polarization of science and post-factual movements, respectively.*

Excerpt of the 4[th] paragraph in Sect. 4:
*Similarly, as suggested by Wooden (2006), the collaboration with local partner institutions (e.g., Gateway Antarctica in New Zealand, the Bjerknes Centre for Climate*

*Research in Norway, the UK Youth Climate Coalition in the UK and Climate Generation in USA) offered experience for successful ways of science communication within each country.*

Excerpts of the last paragraph in Sect. 4:

*Passion united the team and contaminated our audiences, creating better dialogues in a positive feedback loop (Nisbet, 2018).*

*Most importantly, by meeting our audiences in running shoes, on a bicycle or over a beer, we connected as humans - critical to effective science engagement (Nisbet, 2018).*

Excerpt of the last paragraph in Sect. 5:

*When done carefully, we have the potential, regardless of audience's political predilection, to provide trustworthy information to the climate change discourse (Leshner, 2003; MacInnis et al., 2015; Hamilton, 2016).*

These references are:

[revised manuscript text omitted]

not entirely clear how or why two long distance bike rides removes the abstractness of the concept of planetary scale climate change (but I don't really see how you'll change this either - it's a basic premise of the paper that I struggle with). See below comment (and general comment) about a suggested restructure/ reframing, which would remove the need to justify the bike ride in this way.

Based on your suggestion, the relevant excerpt in Sect. 2 now reads:
*The nature of the problem – being a long-term process on a planetary scale – makes it difficult for individuals to grasp and engage with. In an attempt to remove this abstractness, we, as scientists, decided to hit the road in order to share climate science knowledge with people on the ground as well as collect their stories of experienced changes to share them through our platforms. Two journeys from the poles were mapped out: the 10,000-km long bicycle ride - the Southern Cycle - from Christchurch (New Zealand) and the 3,000-km long run - the Northern Run - from Tromsø (Norway), both finishing in Paris during COP 21 (Fig. 1).*

Coincidently vs. in real time
*The public were invited to get behind these journeys and actively become engaged in the climate dialogue ==coincidently==.*

in real time

Based on your suggestion, the relevant excerpt in Sect. 2 now reads:
*The public were invited to get behind these journeys and actively become engaged in the climate dialogue in real time.*

View on COP 21

[revised manuscript text omitted]

37 respondents out of HOW MANY (thousands of) followers? Is this in any way statistically significant or a representative sample of your followers? Surely there's a high level of self-selection and bias in these respondents.

this is in contrast with stats presented above, suggesting that this is not a representative sample

The social media survey had 37 respondents out of more than 6,200 followers on Facebook, 1,200 on Twitter and 650 on Instagram. The number of followers is given in the second last paragraph before the paragraph that this sentence is part of (in Sect. 3). We do neither consider the respondents a significant proportion nor representative sample of our followers, which we also did not claim. Rather, we highlighted the limitations of this survey in the second paragraph following the paragraph that this sentence is part of.

Even with a low sample of our followers (less than 1 %, as written in the manuscript), we chose to include what we saw as the main results of the social media survey in the manuscript. This decision was based on the request from the journal's chief editor Dr. Sam Illingworth. In hindsight, we acknowledge that we should have thought of this earlier, as we would have expected a much higher response rate during the active period of the project than three years after (as discussed in the manuscript).

The 14 answers favouring the actual journeys from the Poles to Paris are indeed not in contrast to the statistics on the Facebook videos presented above this excerpt. Of the four answers (favouring the actual journeys from the Poles to Paris, favouring the physical challenge of running and biking, favouring the scientific message on climate change, and favouring the human face that Pole to Paris put on climate change through stories from the ground), this answer was the least popular, just as the Facebook videos on the journeys from the Poles to Paris were, as discussed in the third last paragraph before the paragraph that this excerpt is part of (in Sect. 3).

*In line with the statistical analysis of the Facebook videos, the fact that ==the scientific message was seen more interesting than the journeys themselves==, indicates that a project like Pole to Paris can find success in disseminating scientific information to a wider audience. Among other key findings from the survey, ==84 % of== the respondents reported that Pole to Paris inspired them in some way. This is also a strong indicator that unconventional projects in the vein of Pole to Paris can find success in connecting with non-scientific audiences in positive ways. ==Moreover, more than half (54 %) indicated that they learned==*

*something new through Pole to Paris, signalling the potential scientists have in bridging the gap between academia and the general public on fundamental societal issues.*

Considering the size of your followers, and the very small response rate, I don't see how you can useful compare response rates of 17 to 14 and read anything into this. Doesn't seem in any way statistically significant.

with only 37 respondents, makes more sense to use actual numbers here ather than a %. Or use both

this sentence is highly problematic. 54% of 37 people is about 20 people. This project would not be seemed a success if it only reached 20 people. Secondly, this is the first time you have mentioned that the cyclists are scientists - and have presented no evidence that it matters that the are scientists. Why do you thnk this response would be different than if it was any young activitists on an adventure?

Please see our reply above.

Following your suggestion, we have replaced percentages by numbers in the relevant excerpt. This now reads:

> *In line with the statistical analysis of the Facebook videos, the fact that the scientific message was seen more interesting than the journeys themselves, indicates that a project like Pole to Paris can find success in disseminating scientific information to a wider audience. Among other key findings from the survey, 31 out of 37 respondents reported that Pole to Paris inspired them in some way. This is also a strong indicator that unconventional projects in the vein of Pole to Paris can find success in connecting with non-scientific audiences in positive ways. Moreover, more than half (20 out of 37) indicated that they learned something new through Pole to Paris, signalling the potential scientists have in bridging the gap between academia and the public on fundamental societal issues.*

Please note that we do not claim Pole to Paris to be a success based on the number of respondents to the social media survey that indicated that they learned something new. Rather, the manuscript aims to discuss what role climate scientists might want to take on in the 21st century. We aim to contribute to this discussion by sharing the experiences from Pole to Paris. In line with the requests of the journal's chief editor, Dr. Sam Illingworth, we use the numbers we have to discuss success without stating that a particular number in itself is an answer to whether or not Pole to Paris was successful. For example, we share numbers on conventional media outlets (close to 250), views of our most popular video (more than 100,000), Twitter tweet reach (almost 250,000) and estimated total reach (more than 1 million) but devote an own section to unquantifiable benefits of Pole to Paris (Sect. 4).

The number of respondents indicating that they learned something new through and that got inspired by Pole to Paris (20 and 31 out of 37, respectively), as we see it, indicate that almost half of our followers already were literate on climate change issues but did not know what to do about it. We managed to inspire them to a large degree. This interpretation is included in a newly written 4th paragraph in Sect. 4 along with a discussion on how we were able to reach people not scientific literate and how our scientific credibility helped in our success. It reads:

*The nature of the Pole to Paris campaign allowed us to build an audience, which did not necessarily have a high interest in science nor necessarily a belief in climate change. This was purposefully done through several means: being on the road and therefore also meeting people who would not otherwise go to a talk about science on climate change; meeting university and school students of all grades and consequently discussing with students who often had barely heard of the science behind climate change; and finally, running and biking, which invited participants for the physical challenge that would stay over for the following talk on climate change and reached by a message they were not initially seeking. This point is also suggested by the number of the social media survey respondents indicating that they learned something new through and that got inspired by Pole to Paris (20 and 31 out of 37, respectively), which indicate that almost half of our followers already were literate on climate change issues but did not know what to do about it. Even though the knowledge and interest in science differ between sociodemographic groups, as suggested by Schäfer et al. (2018), we found that all our audiences had a similar interest in learning about practical actions and solutions they could put in place at a personal level.*

The reference is:
- Schäfer, M. S., Füchslin, T., Metag, J., Kristiansen, S., and Rauchfleisch, A.: The different audiences of science communication: A segmentation analysis of the Swiss population's perceptions of science and their information and media use patterns, Public Underst. Sci., 27, 1–21, doi:10.1177/0963662517752886, 2018.

*Interpreting these numbers, one should keep in mind that the survey respondents already were followers of the climate awareness project Pole to Paris and ==thus not necessarily representative of the average population==. The three year lag of this feedback to the project compared to its most active period also introduce some uncertainty of remembrance and probably explain why ==less than 1 % of our social media followers responded to the survey==. Similarly, the time passed since their publications limited the statistical analysis here to Facebook videos, as other social media data no longer were available. Even so, the ==numbers presented offer valuable insight on the worthiness of time spent on Pole to Paris and can help the outreach community in learning from our efforts==.*

or of the followers... they're not representative of anything!

don't think the survey has any validity unless you can in some way show it is representative of your followers, which you haven't

really? from 20 people?

Please note that due to the large span of target group, considering geography, age and backgrounds, of Pole to Paris, it is very hard to say what a typical follower looks like. We have also addressed this indirectly throughout the manuscript when talking about the reach of the project to audiences in different countries, speaking different languages, in different media and with different cultural and educational backgrounds. We hope that these are constructive answers to your two first claims.

The third claim or question is unfortunately missing our intended meaning. We hope that we have clarified what we mean with the highlighted excerpt above, which does not refer to the

fact that 20 respondents learned something new through Pole to Paris, but rather to all other numbers presented from Pole to Paris at this point and the added benefits - for us, our audiences and the scientific community - to be presented in the section that follows.

Based on your suggestions, we have rewritten the relevant excerpt in Sect. 3. It now reads:

*Interpreting these numbers, one should keep in mind that the survey respondents already were followers of the climate awareness project Pole to Paris and thus not necessarily representative of the average population. The three-year lag of this feedback to the project compared to its most active period also introduce some uncertainty of remembrance and probably explain why less than 1 % of our social media followers responded to the survey. This small respondent rate meant that the answers not necessarily represented those of a typical follower. Moreover, the time passed since most of our social media posts were published limited the statistical analysis here to Facebook videos, as other social media data no longer were available. Even so, we believe the numbers presented in this manuscript offer valuable insight on the worthiness of time spent on Pole to Paris and can help the outreach community in learning from our efforts.*

Two-way interaction
*As also mentioned by Barnosky et al. (2014), the direct success of an initiative like Pole to Paris is however almost impossible to quantify. Indirectly, the Pole to Paris team members took great value from being able to deliver vital information to the public. Engaging in two-way interaction with a range of audiences – from farmers to senators, from preschool children to retirees and from Norwegians to Bangladeshis – provided invaluable insight to our own research questions. Fortunate with these encounters, we faced questions and concerns often far from ours, which opened our eyes and ears and widened our perspectives.*

This is more about the value the experience gave to the team members than the people they spoke with - which is not to be underestimated. "Two-way" interaction implies genuine two-way dialogue, but this sounds more like it was the adventurer-scientists telling the people they met what they knew about climate change... so more like education than 2-way interaction.

We acknowledge that this excerpt was written poorly, making it sound that we were delivering information to our audiences without a two-way communication. This was far from the truth. We did indeed share knowledge of climate science and experiences from the road to Paris with our audience, but we equally listened to our audiences' experiences of climate change and answered their questions on climate science. By rephrasing the relevant wording on our dialogue approach, we hope and believe that we have been able to better highlight this vital part of Pole to Paris in the updated manuscript.

Based on your suggestion, the 2nd–3rd sentence in the 2nd paragraph in Sect. 2 now reads:
*The nature of the problem – being a long-term process on a planetary scale – makes it difficult for individuals to grasp and engage with. In an attempt to remove this abstractness, we, as scientists, decided to hit the road in order to share climate science knowledge with people on the ground as well as collect their experienced changes and share them through our platforms.*

Based on your suggestion, the relevant excerpt in Sect. 4 now reads:

*As also mentioned by Barnosky et al. (2014), the direct success of an initiative like Pole to Paris is however almost impossible to quantify. Indirectly, the Pole to Paris team members took great value from being able to* *share climate science with our audiences and listen to their experiences of climate change.* *Engaging in two-way interaction with a range of audiences – from farmers to senators, from preschool children to retirees and from Norwegians to Bangladeshis – provided invaluable insight to our own research questions. Fortunate with these encounters, we faced questions and concerns often far from ours, which opened our eyes and ears and widened our perspectives.*

The Paris Agreement as a success factor

The Paris Agreement, of which Pole to Paris was one of many numerous initiatives building public support for, was arguably a better outcome of COP21 than the climate science community could have hoped for.

This is problematic as well - while I don't disagree with this sentence, I don't quite get the connection between this sentence and the previous paragraph (which makes an important point about the immeasurable, multiple and qualititative benefit of the journey).

Based on your suggestion, the relevant excerpt in Sect. 4 now reads:

*The Paris Agreement, of which Pole to Paris was one of numerous initiatives building public support for, was arguably a better outcome of COP21 than the climate science community could have hoped for* *and, as later similarly suggested by Drummond et al. (2018), might have been influenced by that awareness raised among people.*

The reference is:

- Drummond, A., Hall, L. C., Sauer, J. D. and Palmer, M. A.: Is public awareness and perceived threat of climate change associated with governmental mitigation targets?, Climatic Change, 149, 1–13, doi:10.1007/s10584-018-2230-2, 2018.

Critization and credibility of science communicators

*Schimid and Petri (2017) have argued that those in the scientific community who actively attempt to communicate the seriousness of climate change to a wide audience often are met with attempts "to discredit their scientific credibility, or to criticize the studies that are used or their underlying methods and models." As communicators of the scientific consensus, we inevitably experienced these tactics from climate sceptics in online fora. Mostly, the criticisms were from individual citizens and directed at us personally. Out on the roads to Paris, however, fact-based messaging was immensely welcomed. Considering the politicized division of the media themselves (e.g., Brüggemann and Engesser, 2017), this positive experience of direct engagement supports the suggestion by Gauchat et al. (2017) that science participation and outreach could rebuild the credibility among communities most critical of scientists.*

This is a nice, and important, paragraph

Thank you!

Professional credibility and climate change communication

*Hence, we worked hard to keep our credibility as researchers (Nordhagen et al., 2014), not partnering with organizations or initiatives on either of the climate advocacy fringes, and not favouring one political party over another. We experienced a boost in personal and public credibility, more than outweighing a loss in ==professional credibility from our publication record hiatus while on the road==, thus overall enhancing our researcher credibility. We saw our role as awareness-raisers, increasing the understanding of climate science within all societal groups.*

the loss of professional credibility associated with climate change communication is not usually about a reduced number of publications, but rather the apparent politicisation of "objective" scientists.

In writing this excerpt, we used the definitions and discussions in Nordhagen et al. (2014) on a researcher's credibility, which they separate into professional, public and personal credibilities. There, they define professional credibility as the credibility a researcher has across academic (and non-academic) research communities of various scales, public credibility as the credibility a researcher has in the wider population (including policymakers, media and "lay people") and personal credibility as the credibility a researcher has from relationships with self, family, friends and close associates.

According to Nordhagen et al. (2014), "[a]n academic researcher gains professional credibility by obtaining academic qualifications and adhering to common research principles [...] such as observing ethical standards (e.g., acknowledging funding, avoiding interest conflicts), publishing peer-reviewed research, methodological transparency, and data availability". Under public credibility, they write that "[t]here is a difference between engaging on issues of science and those of policy; one must not conflate views of 'things done in the name of science' with views of science itself [...]. While scientific papers and grants commonly emphasise policy-relevance and include policy recommendations, these rarely translate into policy-making involvement. Indeed many argue this ought be avoided as potentially damaging to the researcher's professional and public credibility." Furthermore, for the personal credibility of a climate researcher, they write that "aspects of their personal behaviour (and associated carbon footprint) cannot be viewed in isolation from their professional expertise."

Nordhagen et al. (2014) then go on to discuss the conflict between professional and public credibilities: "Often drawn to climate change research by personal interest and belief in the necessity of curbing emissions, researchers may find themselves disagreeing with particular government actions/inactions and feel compelled to join public calls for stronger action. This contrasts with the common academic view that professional credibility demands researchers set aside their citizens' rights/responsibilities to hold the government accountable on their area of expertise. [...] This could, by some arguments [...], damage the overall scientific credibility of climate change research if researchers are seen as having an agenda. Yet one can also argue that while the researcher should not undermine scientific credibility of research, entering the profession does not annul their right to petition/protest: concern over 'credibility as a scientist' should not cause the scientist to 'disregard his credibility as a human being and voter with genuine convictions'".

Hence, as we see it, we could have written much more on the topic in the light of the discussion in Nordhagen et al. (2014). However, considering that we do not wish to lengthen our manuscript more than strictly necessary, we originally kept this discussion short by referencing Nordhagen et al. (2014). Nevertheless, to accommodate the important point you brought up, we have added a bit more discussion on the topic in the updated manuscript.

Hence, based on your suggestion, the relevant excerpt in Sect. 4 now reads:

*Consequently, we worked hard to keep our credibility as researchers (Nordhagen et al., 2014), not partnering with organizations or initiatives on either of the climate advocacy fringes, and not favouring one political party over another. Based on the feedback received, this scientific background and endeavour to remain objective allowed us to partner with organisations otherwise not within reach, like the United Nations Development Programme (UNDP) and the World Meteorological Organization. Following the definitions by Nordhagen et al. (2014), we experienced a boost in personal and public credibility, more than outweighing a loss in professional credibility from our publication record hiatus while on the road, thus overall enhancing our researcher credibility. By being open about what role we played in public, we strove to negotiate the tension between our professional and public credibilities discussed by Nordhagen et al. (2014), in which our goal of stronger climate action on governmental level to some degree was challenged by the common academic view that researchers should remain detached from public policies. We saw our role as awareness-raisers, increasing the understanding of climate science within all societal groups.*

Framing of climate change
*Spanning the cultural differences within these groups, we tailored the message to the audiences in line with the suggestions by Somerville and Hassol (2011). These included framing climate change as a human and not only an environmental issue, focusing on the now instead of the decades ahead, leading with what we know, using a language adapted to a public discourse, being passionate, and connecting the dots between climate change and the personal experiences of the audience themselves.*

this is nice.

Thank you!

Backgrounds in and set-up of the Pole to Paris team
*While having a global focus, the ten languages spoken by the highly international group members allowed us to personally engage with a wide range of people on the road from the polar regions to Paris.*

Until this point in the paper, I had assumed there were two people biking, on two different routes. Now it seems there were several people on the journeys - in which case much earlier in the piece you need to explain the absolute basics about the journey - who, when, how many people, what they do in their day jobs (since this apparrently becomes relevant later on), age, demographic, nationality etc.

Thank you for pointing out that the backgrounds and set-up of the Pole to Paris team was not clear before this excerpt (in Sect. 4). For this reason, we have added more information about the Pole to Paris team in the beginning of Sect. 2.

Please note that just one of the two routes was cycled. The 3,000-km long Northern Run was run. We hope that the added text on the Pole to Paris team clarifies the journeys in Sect. 2 and Fig. 1.

Hence, based on your suggestion, the 1st–3rd paragraphs in Sect. 2 now read:

[revised manuscript text omitted]

what does this mean? Only the cyclists participated physically, surely all other followers/audiences followed virtually or through in-person interaction along the way

Thank you for pointing out this ambiguity. Based on your suggestion, the relevant excerpt in Sect. 4 now reads:
*Besides, these language skills helped spread our messages even further, as suggested by the 62 % followers on Facebook speaking English, 16 % Indonesian, 6 % Norwegian, 4 % French, 3 % Spanish and 2 % German.*

Local partner institutions and Global Voices
*Similarly, the collaboration with local partner institutions offered tools of experience for successful ways of science communication within each country. This collaboration also allowed us to organize Global Voices events outside of the Northern Run and Southern Cycle (Fig. 1), during which youth came together to learn about climate change and how they could act upon it.*

can you say who this is?

Is this seperate from the bike ride? I think a focus on both the bike rides AND these events could be really interesting as a "case study" for how scientists were trying to raise awareness in the lead up to the Paris meeting.

Based on your suggestions, the relevant excerpt in Sect. 4 now reads:
*Similarly, as suggested by Wooden (2006), the collaboration with local partner institutions (e.g., Gateway Antarctica in New Zealand, the Bjerknes Centre for Climate Research in Norway, the UK Youth Climate Coalition in the UK and Climate Generation in USA) offered experience for successful ways of science communication within each country. This collaboration also allowed us to organize what we called the Global Voices events with our partner UNDP. These were set up outside the routes of the Northern Run and Southern Cycle (Fig. 1), during which youth came together to learn about climate change and how they could act upon it.*

Sentence on other climate change awareness campaigns
*The experiences from Pole to Paris were, however, not unique. Other initiatives have been launched over the* ==lasts years to increase awareness on climate changes and train scientists in science communication==.

check grammar, spelling and sentence structure for intended meaning.

Based on your suggestion, the relevant excerpt in Sect. 4 now reads:
> *The experiences from Pole to Paris were, however, not unique. Other initiatives have been launched over the last* few *years to increase climate change* awareness *and train scientists in* more effective *science communication.*

Reference to Editorial (2017)
*However, March for Science has also been criticized, as it runs the risk of creating a false picture of scientists being more driven by ideology than evidence (*==Editorial, 2017==*).*

check this is correctly referenced

Thank you for pointing out this erroneous citation. This reference refers to the editorial in Nature on April 11, 2017 (https://www.nature.com/news/nature-supports-the-march-for-science-1.21804). Unfortunately, no author is listed. As far as we understand, the first couple of words in the title of the editorial should then be given as an in-text citation. In the biography, we have added the day and month too.

Hence, based on your suggestion, the relevant excerpt in Sect. 4 now reads:
> *However, March for Science has also been criticized, as it runs the risk of creating a false picture of scientists being more driven by ideology than evidence (*Nature *supports the March for Science, 2017).*

Similarly, the relevant reference in the biography now reads:
> Nature *supports the March for Science, [Editorial], Nature, 544, 137,* doi:10.1038/544137a, 11 April *2017.*

Reference to Will You Hear Us
*Furthermore, the authors have been involved in other more or less politically charged outreach projects. Climate Communication Cologne is an effort launched at the University of Cologne whose main objective is to facilitate science communication to a wide non-academic audience. This takes place in various forms, such as workshops, stand-up comedy or videos, and in various arenas, from schools and universities to pubs and online communities.* ==Will You Hear Us *is a documentary on the tradition of caged birds in Indonesia, which has become unsustainable due to the ever-increasing demand for wild songbirds and poses a huge threat on biodiversity*==.

On its own like this, this sentence is a bit of a non-sequitur. However, this is becoming a much more interesting story when you present Pole to Paris as one of a number of outreach initiatives you are involved in - to me, that is a much more interesting story to tell, keep it autobiographical/ auto-ethnographic, and use your supporting data to help illustrate the

experience rather than suggesting it is particularly compelling data/ evidence on its own. It's VERY hard to measure impact, but very useful to tell a story.

Thank you for pointing out the little coherence in this paragraph. Based on your suggestion, the relevant excerpt in Sect. 4 now reads:

*Furthermore, the authors have been involved in other more or less politically charged outreach projects. For instance, Climate Communication Cologne is an effort launched at the University of Cologne whose main objective is to facilitate science communication to a wide non-academic audience. This takes place in various forms, such as workshops, stand-up comedy or videos, and in various arenas, from schools and universities to pubs and online communities. Another example is Will You Hear Us, a documentary on the tradition of caged birds in Indonesia, which has become unsustainable due to the ever-increasing demand for wild songbirds and poses a huge threat on biodiversity. Both authors are currently also writing comic books on climate change adaptation and mitigation and on biodiversity loss for high-school and elementary school students, respectively.*

Reference to Pielke Jr. (2007)
*In our current society, we argue that the role of the ‘pure scientist’ (as defined by Rapley and De Meyer, 2014) is outdated and the need of the ‘science communicator’ is rising.*

worth also reading Pielke: The Honest Broker

Based on your suggestion, the relevant excerpt in Sect. 5 now reads:

*In our current society, we argue that the role of the ‘pure scientist’ (as defined by Rapley and De Meyer, 2014) is outdated and the need of the ‘science communicator’ and ‘the honest broker of policy alternatives’ (as outlined by Pielke Jr., 2007) is rising.*

The advancement of science
*The advancement of science is completely pointless if it is ignored by government as well as the general public and not suitably utilised by an educated society.*

Be careful here - are you specifically talking about science related to climate change? If so, keep that clear... otherwise you're somewhat arrogantly dismissing all blue skies "pure science" in completely different disciplines.

Based on your suggestion, the relevant excerpt in Sect. 5 now reads:

*The advancement of science might be of little significance if it is ignored by government as well as the laypeople and not suitably utilised by an educated society.*

Grammar on the benefits of engaging in outreach activities
*For scientists at the beginning of their academic career, we support the notion by Brownell et al. (2013) and Rauser et al. (2017) that engaging in outreach activities helps shaping the research questions, giving more effective tools for narrowing the widening gap between academia and the general public, and eventually providing a more constructive input for policy formulation on climate change.*

check grammar

Based on your suggestion, the relevant excerpt in Sect. 5 now reads:

*For scientists at the beginning of their academic career, we support the notion by Leshner (2007), Brownell et al. (2013), Rauser et al. (2017) and Nisbet (2018) that engaging in outreach activities helps shape the research questions, giving more effective tools for narrowing the widening gap between academia and the rest of society, and eventually providing a more constructive input for policy formulation on climate change.*

Adapting and playing our new role as climate scientists
*Now it is up to us to adapt and play our new role objectively while keeping our credibility.*

This is a loaded sentence, probably best avoided or edited unless you're going to expand on what that would look like - to be objective, retain credibility, and adapt to the political climate. Probably too complex for this short commentary.

Based on your suggestion and referring to the excerpts from Nordhagen et al. (2014) in our reply to Professional credibility and climate change communication above, we have added a sentence after the highlighted sentence. The relevant excerpt in Sect. 5 now reads:

[revised manuscript text omitted]

Slettet: commentary

Slettet: .

Slettet: other

Slettet: undergird

Formatert: Hevet

Formatert: Hevet

Slettet:  and

Slettet: .

Slettet: galvanize public

awareness of climate change in a country is positively related to the unconditional climate mitigation targets of that country, as later suggested by Drummond et al. (2018).

The Pole to Paris project focused on reshaping the way scientists engage with the public on climate change issues. The nature of the problem – being a long-term process on a planetary scale – makes it difficult for individuals to grasp and engage with. In an attempt to remove this abstractness, we, as scientists, decided to hit the road in order to share climate science knowledge with people on the ground as well as collect their stories of experienced changes to share them through our platforms. Two journeys from the poles were mapped out: the 10,000-km long bicycle ride – the Southern Cycle – from Christchurch (New Zealand) and the 3,000-km long run – the Northern Run – from Tromsø (Norway), both finishing in Paris during COP 21 (Fig. 1). These journeys were led by two climate scientists, who left Christchurch and Tromsø shortly after completing their PhDs in Antarctic and Arctic climate change, respectively. 7.5 and 4 months later, respectively, they reached Paris. They were supported by the eight other Pole to Paris team members, whose backgrounds ranged from environmental and political science to web and product design. While all members actively contributed to Pole to Paris by various means from their locations around the world, five of them also joined the main cyclist and runner for part of the journeys. Of the ten team members, only the main cyclist and runner were working full-time on the project (i.e., without getting paid), while the others had studies or jobs to balance simultaneously. Whereas we were all in our 20s, the four female and six male team members represented eight different countries.

The public were invited to get behind the Southern Cycle and Northern Run journeys and actively become engaged in the climate dialogue in real time. The adventure component also helped to attract media attention, giving the project a platform to communicate the facts about climate change and the importance of COP 21 to the wider audience by engaging them in the journeys. Crucially, along the way, we held talks in schools, universities and many other public venues and were joined by other cyclists and runners for part of the distances. This created a two-way communication 
[revised manuscript text omitted]

**Formatert:** Innrykk: Venstre: 1,27 cm, Ingen punktmerking eller nummerering

**Slettet:** ↵

**Formatert:** Innrykk: Venstre: 1,27 cm, Ingen punktmerking eller nummerering

**Slettet:** ↵

**Formatert:** Innrykk: Venstre: 1,27 cm, Ingen punktmerking eller nummerering

**Slettet:** ↵

**Formatert:** Innrykk: Venstre: 1,27 cm, Ingen punktmerking eller nummerering

**Slettet:** ↵

**Formatert:** Innrykk: Venstre: 1,27 cm, Ingen punktmerking eller nummerering

**Slettet:** 'pure scientist'

**Slettet:** 'science communicator'

**Slettet:** is completely pointless

**Slettet:** general public

**Slettet:** .

**Slettet:** ) and

**Slettet:** helps shaping

**Slettet:** general public

**Slettet:** this

**Slettet:** .

**Slettet:** audience's

**Slettet:** "

**Slettet:** "

[revised manuscript text omitted]

**Slettet:** Won't Do It"

**Slettet:** Positive Engagement With Climate Change Through Visualpositive engagement with climate

**Slettet:** Iconic Representations

Panel on Climate Change, Intergovernmental Panel on Climate Change (IPCC), Cambridge, UK and New York, USA, 2013.

Stoknes, P. E.: What we think about when we try not to think about global warming: Toward a new psychology of climate action, Chelsea Green Publishing, Vermont, USA, ISBN:978-1-60358-583-5, 2015.

Theel, S., Greenberg, M., and Robbins, D.: Study: media sowed doubt in coverage of UN climate report, Media Matters for America, https://www.mediamatters.org/research/2013/10/10/study-media-sowed-doubt-in-coverage-of-un-clima/196387, 2013.

Townshend, T., Fankhauser, S., Aybar, R., Collins, M., Landesman, T., Nachmany, M., and Pavese, C.: How national legislation can help to solve climate change, Nat. Clim. Change, 3, 430–432, doi:10.1038/nclimate1894, 2013.

Townson, S.: Why people fall for pseudoscience (and how academics can fight back), The Guardian, https://www.theguardian.com/higher-education-network/2016/jan/26/why-people-fall-for-pseudoscience-and-how-academics-can-fight-back, 2016.

Ungar, S.: The rise and (relative) decline of global warming as a social problem, Sociol. Q., 33, 483–501, doi:10.1111/j.1533-8525.1992.tb00139.x, 2016.

Vraga, E., Myers, T., Kotcher, J., Beall, L., and Maibach, E.: Scientific risk communication about controversial issues influences public perceptions of scientists' political orientations and credibility, Roy. Soc. Open Sci., 5, 170505, doi:10.1098/rsos.170505, 2018.

Witte, K.: Putting the fear back into fear appeals: The extended parallel process model, Commun. Monogr., 59, 329–349, doi:10.1080/03637759209376276, 1992.

Wooden, R.: The principles of public engagement: at the nexus of science, public policy influence, and citizen education, Soc. Res., 73, 1057–1063, 2006.

Zhao, X., Rolfe-Redding, J., and Kotcher, J. E., Partisan differences in the relationship between newspaper coverage and concern over global warming, Public Underst. Sci., 25, 543–559, doi:10.1177/0963662514558992, 2016.

Zürn, M.: The politicization of world politics and its effects: Eight propositions, Eur. Polit. Sci. Rev., 6, 47–71, doi:10.1017/S1755773912000276, 2014.

[Figure]

**Figure 1: Map of the two Pole to Paris journeys: the Northern Run (blue trajectory) and the Southern Cycle (red trajectory); as well as the Global Voices events organized in collaboration with partners (green dots).**

[Figure]

**Figure 2: Percentages of total Facebook video views after three seconds (>3 s), at 30 seconds (or to the end, whichever came first; ≧30 s), and at 95 % of the video length (including people that skipped to this point; 95 %) for (a) organic (i.e., not paid; blue columns) and paid (red columns) views and (b) videos on climate action (blue columns), climate change (red columns) and the journeys themselves (green columns). Numbers above the columns in (a) and (b) represent total and average views, respectively.**

---

## Author Comment (AC2) · 7 Feb 2019

Dear MSc. Kristin Timm,

We thank you for your suggestions on the manuscript. With the changes explained below, we feel that the paper is strengthened compared to its first submission.

In the attached document, we have gone through each comment by the reviewers and explained our choices of changes in accordance with these. This document also includes the tracked changes of the updated manuscript compared to its first version.

Yours sincerely, Dr. Erlend Moster Knudsen and MSc. Oria Jamar de Bolsée

Please also note the supplement to this comment:

[Figure]

https://www.geosci-commun-discuss.net/gc-2018-16/gc-2018-16-AC2-supplement.pdf

**[GCD](https://doi.org/10.5194/gc-2018-16)**

Interactive
comment

---

## Author Comment (AC3) · 7 Feb 2019

**The role of climate scientists in the post-factual society**

Erlend M. Knudsen and Oria J. de Bolsée

**Set-up of response**

We thank the reviewer for her suggestions on the manuscript. With the changes explained below, we feel that the paper is strengthened compared to its first submission.

In the following, we go through each comment by the reviewers (reproduced here in gray text for your reference) and explain our choices of changes in accordance with these. Where changes to the text in the manuscript are made, the relevant excerpt is reproduced from the .pdf manuscript to this .docx response in *italic text*, with changes written in *italic green text*.

**Response to the review by Kristin Timm**

Reasons for climate action message success

The most important finding in this paper, in my opinion, is that the climate action message - coming from climate scientists - was the most popular (among social media video viewers). This finding is not surprising, considering communication theories like the Extended Parallel Processing Model (Witte) suggest that increasing efficacy to cope with risk is a critical part of effective risk message processing. However, this finding (or reflection) is most interesting to me considering that many climate scientists are hesitant to venture past explaining the causes and effects of climate change, into discussing solutions, for fear of being seen as an advocate and losing credibility. The authors could strengthen this paper by spending more time reflecting on and discussing the content of their climate action messages and exploring where those messages sit on a spectrum from objective to advocate. Furthermore, these reflections could be more strongly placed in the existing literature about the role of scientists in society.

Thank you for sharing your views on the manuscript. We agree that reflecting on and discussing how messages of climate action is seen in terms of climate advocacy is an important aspect following our experiences. For this reason, we have made several changes to the manuscript.

In particular, the end of the 3rd paragraph in Sect. 2 now reads:

*In line with O'Neill and Nicholson-Cole (2009) and Stoknes (2015), we highlighted the opportunities and inspiration of acting on climate change now rather than later. We communicated the ongoing and expected consequences of climate change, but in terms of relevant and experienced changes rather than fear rising from their cognitive dissonance following Extended Parallel Processing Model theory (Witte, 1992).*

Similarly, we have added a sentence on this matter in the 2nd paragraph in Sect. 4, including two references:

*Moreover, fostering constructive public conversations about science and society can, among others, improve decision-making, promote trust and credibility in scientific findings and strengthen democratic processes (Wooden, 2006; Nisbet, 2018), ultimately*

*counteracting politicization and polarization of science and post-factual movements, respectively.*

We have also expanded the 3rd paragraph in Sect. 4, which discusses the credibility of researchers. This now reads:

*Consequently, we worked hard to keep our credibility as researchers (Nordhagen et al., 2014), not partnering with organizations or initiatives on either of the climate advocacy fringes, and not favouring one political party over another. Based on the feedback received, this scientific background and endeavour to remain objective allowed us to partner with organisations otherwise not within reach, like the United Nations Development Programme (UNDP) and the World Meteorological Organization. Following the definitions by Nordhagen et al. (2014), we experienced a boost in personal and public credibility, more than outweighing a loss in professional credibility from our publication record hiatus while on the road, thus overall enhancing our researcher credibility. By being open about what role we played in public, we strove to negotiate the tension between our professional and public credibilities discussed by Nordhagen et al. (2014), in which our goal of stronger climate action on governmental level to some degree was challenged by the common academic view that researchers should remain detached from public policies. We saw our role as awareness-raisers, increasing the understanding of climate science within all societal groups.*

Moreover, we have expanded the first sentence in the 2nd paragraph in Sect. 5 with another relevant reference. This now reads:

*In our current society, we argue that the role of the 'pure scientist' (as defined by Rapley and De Meyer, 2014) is outdated and the need of the 'science communicator' and 'the honest broker of policy alternatives' (as outlined by Pielke Jr., 2007) is rising.*

Furthermore, we have expanded the 3rd paragraph in Sect. 5, also with more references, which now reads:

*For scientists at the beginning of their academic career, we support the notion by Leshner (2007), Brownell et al. (2013), Rauser et al. (2017) and Nisbet (2018) that engaging in outreach activities helps shape the research questions, giving more effective tools for narrowing the widening gap between academia and the rest of society, and eventually providing a more constructive input for policy formulation on climate change. As we see it, this will act to reduce politicization and polarization of climate change, while also depressing the breeding ground for post-factual movements. Within academia, outreach training gives us better tools in teaching, mentoring of younger students and taking part in scientific discussions, as well as contributing to better written research proposals and journal publications (Stiller-Reeve et al., 2016, and references therein).*

Finally, we have added two more references to the relevant sentences in the last paragraph in Sect. 5 on this topic, which now read:

*Now it is up to us to adapt and play our new role objectively while keeping our credibility (as discussed by Nordhagen et al., 2014). According to Rapley and De Meyer (2014), this has the potential to remove climate science from the direct firing line to leave the authority, responsibility and accountability for decisions transparently with the policymakers and the public. When done carefully, we have the potential, regardless of audience's political*

*predilection, to provide trustworthy information to the climate change discourse (Leshner, 2003; MacInnis et al., 2015; Hamilton, 2016).*

These references are:

- Leshner, A. I.: Public engagement with science, Science, 299, 977, doi: 10.1126/science.299.5609.977, 2003.
- Leshner, A. I.: Outreach training needed, Science, 315, 161, doi:10.1126/science.1138712, 2007.
- MacInnis, B.,Krosnick, J. A., Abeles, A., Caldwell, M. R., Prahler, E., and Dunne, D. D.: The American public's preference for preparation for the possible effects of global warming: impact of communication strategies, Climatic change, 128, 17–33, doi:10.1007/s10584-014-1286-x, 2015.
- Nisbet, M.: Scientists in civic life: facilitating dialogue-based communication, American Association for the Advancement of Science, https://www.aaas.org/sites/default/files/s3fs-public/content_files/Scientists%2520in%2520Civic%2520Life_FINAL%2520INTERACTIVE%2520082718.pdf, 2018.
- Pielke Jr., R. A.: The honest broker: making sense of science in policy and politics, Cambridge University Press, New York, USA, ISBN:978-0-521-87320-8, 2007
- Stoknes, P. E.: What we think about when we try not to think about global warming: Toward a new psychology of climate action, Chelsea Green Publishing, Vermont, USA, ISBN:978-1-60358-583-5, 2015.
- Witte, K.: Putting the fear back into fear appeals: The extended parallel process model, Commun. Monogr., 59, 329–349, doi:10.1080/03637759209376276, 1992.
- Wooden, R.: The principles of public engagement: at the nexus of science, public policy influence, and citizen education, Soc. Res., 73, 1057–1063, 2006.

**Historical development of the climate scientist role**

The authors say on page 1, line 13, "The role of climate science in the public sphere has changed significantly since the mid-1980s." I would like to hear more about this line of reasoning, and I recommend the authors explore some of the existing literature from science and technology studies that reflects on the role of scientists in society (i.e. The Honest Broker, by Roger Peilke Jr., recent work by John Kotcher et al. also explores scientists' advocacy messages).

Based on your suggestion, we have expanded the 1st paragraph in Sect. 1 into three. These include references to "The honest broker" by Roger A. Pielke Jr. (Pielke Jr., 2007) and two of the recent works by John Kotcher (Vraga et al., 2018 and Zhao et al., 2016) and read:

*The role of climate science in the public sphere has changed significantly since the mid-1980s. Ensuing the formation of the Intergovernmental Panel on Climate Change (IPCC) and the U.S. Senate testimony of James Hansen in 1988, climate science has increasingly become a topic of political debate, media coverage and part of the daily discourse in our societies (Bolin, 2007; Ungar, 2016). Simultaneously, the scientific understanding of climate change has been rapidly expanding, with the number of climate change papers published per year exponentially growing (McSweeney, 2015) and the confidence in humans as the main cause of global warming has gone from insufficient to "extremely likely" (as defined by the IPCC First to Fifth Assessment Reports; Houghton et al., 1990; Stocker et al., 2013).*

*A corresponding increase has neither been seen in climate change legislation (Townshend et al., 2013), media coverage of climate change topics (Boykoff et al., 2018) nor in public perception of climate change (Capstick, et al., 2015; Zhao et al., 2016; Saad, 2017). Instead, the politicization and polarization of climate change has been growing, with the former referring to how the science behind political decisions increasingly are promoted and attacked by advocates and opponents and the latter referring to the growing division between elites, organisations and political parties viewing climate change as a negative consequence of industrial capitalism and those opposing such views (McCright and Dunlap, 2011). This trend is arguably most notable in the U.S. (Capstick et al., 2015; Carmichael et al., 2017), where the partisan divide on environmental voting score (as defined by the League of Conservation Voters) grew from about 25 in 1970 to about 85 in 2015 (Dunlap et al., 2016). Since then, Donald Trump was elected as the country's 45th president and has repeatedly been questioning climate science, actively working against environmental legislation and funding of his predecessor and generally making the work of climate scientists more challenging (De Pryck and Gemenne, 2017; Alderman and Inwood, 2018; and references therein). A post-factual society has arisen, in which part of its members rather accept an argument based on their emotions and beliefs than one based on scientific facts (Alvermann, 2017).*

*A post-factual political scene is not isolated to the U.S. alone; Brexit in the U.K. and the (re-)elections of Rodrigo Duterte in the Philippines, Andrzej Duda in Poland, Viktor Orbán in Hungary, Recep Tayyip Erdogan in Turkey and Jair Bolsonaro in Brazil are all examples of populistic solutions trumping science-based ones (Postel-Vinay, 2017). Furthermore, the rise of social media has meant that everyone can act as journalists and editors in choosing what to post, where algorithms make sure to share posts from those with similar opinions, thus creating filter bubbles (Pariser, 2011; Alvermann, 2017; Bail, 2018). Conventional media can also reinforce filter bubbles by presenting scientific news within pre-existing worldviews of their audiences (Theel et al., 2013; Carmichael et al., 2017). Similar bubbles exist within academia, where scientists are trained to write for an already highly educated and specialized audience (Stiller-Reeve et al., 2016). Scientists are thus often seen as an elite without touch to the rest of society (Townson, 2016). For this reason, it is, more than ever, crucial to establish dialogues with those outside of academia in order to help trigger positive global changes (Leshner, 2007; Barnosky et al., 2016). Doing so, we, as scientists, need to choose our role within society carefully in consideration of the consequences for us individually and as a community (Pielke Jr., 2007; Vraga et al., 2018).*

Data transparency
While I appreciate the presentation of the data related to the campaign, I encourage the authors to be much more transparent about who exactly was engaged in the different elements of the campaign and where they have data and where they do not. On page 2, line 23, the authors said, "A conservative estimation is that more than one million people in 45 countries were reached through conventional and social media." If it is included, I would like to see a much more detailed description of how this figure was produced. Is this based on social media impressions? Is this based on traditional media circulation rates? The analysis of the social media videos is interesting, but the authors should acknowledge the extent to which this audience is similar to or distinct from the audiences who participated in public lectures and those who engaged with the campaign through traditional news sources and the population as a whole.

To that end, I would strongly recommend the authors avoid the use of the term "general public." From the description provided, several distinct audiences were targeted and reached during the campaign (i.e. school children, people who attended a lecture, people who watched a video on Facebook) - and each of these audiences likely has unique characteristics that are relevant when considering the authors' final outreach recommendations. In particular, I encourage the authors to address the extent to which their campaign attracted people who already had a high interest in science or belief in climate change (see Besley, "Audiences for Science Communication" for further discussion from a US context).

Thank you for highlighting that we were not transparent enough in the manuscript. We always wished to be so, so that others can learn from our experiences. For this reason, we have tried following your suggestions and edited the text accordingly.

Hence, based on your suggestion, we elaborated on the more than one million people estimate, making it an own paragraph at the end of Sect. 2. This paragraph also touches upon how our followers differed between conventional and social media. It reads:

*A conservative estimation is that more than one million people in 45 countries were reached through conventional and social media, which included close to 250 media outlets and almost 500,000 and 250,000 reached per Facebook post and Twitter tweet, respectively. While it is probable that some of our followers on Facebook, Twitter and Instagram overlapped, the breadth of conventional media coverage meant that we were able to reach a wider span of the society. For example, our story was featured five times on CNN in English, Spanish and Arabic, while Norwegian Broadcasting Corporation aired us 14 times. None of these are likely to be seen by the average Thai, Chinese or Indonesian, but our appearance in the Thai news channel TNN24, the China News Service or the Indonesian Jawa Pos might. Similarly, where coverage in the English-language news actors The Guardian, The Huffington Post or The Daily Star plausibly caught the attention of those already aware of human-induced climate change, the more domestic-focused Le Parisien in French, la Repubblica in Italian or Correio Braziliense in Portuguese almost certainly brought climate change into new light among their readers. Additionally, we gave 80 presentations in five languages along the running route alone.*

Moreover, we have added a new 4th paragraph in Sect. 4, which discusses how our audiences differed among our communication forms. In this discussion, we referenced Schäfer et al. (2018) instead of Besley (2018) due to balance the already relative high number of studies from the U.S. with more from Europe, where a larger part of Pole to Paris took place. The paragraph reads:

*The nature of the Pole to Paris campaign allowed us to build an audience, which did not necessarily have a high interest in science nor necessarily a belief in climate change. This was purposefully done through several means: being on the road and therefore also meeting people who would not otherwise go to a talk about science on climate change; meeting university and school students of all grades and consequently discussing with students who often had barely heard of the science behind climate change; and finally, running and biking, which invited participants for the physical challenge that would stay over for the following talk on climate change and reached by a message they were not initially seeking. This point is also suggested by the number of the social media survey respondents indicating that they learned something new through and that got inspired by Pole to Paris (20*

*and 31 out of 37, respectively), which indicate that almost half of our followers already were literate on climate change issues but did not know what to do about it. Even though the knowledge and interest in science differ between sociodemographic groups, as suggested by Schäfer et al. (2018), we found that all our audiences had a similar interest in learning about practical actions and solutions they could put in place at a personal level.*

The reference is:

- Schäfer, M. S., Füchslin, T., Metag, J., Kristiansen, S., and Rauchfleisch, A.: The different audiences of science communication: A segmentation analysis of the Swiss population's perceptions of science and their information and media use patterns, Public Underst. Sci., 27, 1–21, doi:10.1177/0963662517752886, 2018.

Finally, based on your suggestion, we have replaced the term "general public" with more specific terms for the meanings intended throughout the manuscript. The sentences, which they belong to, are reproduced here.

Excerpt of the Abstract:

*We share our experiences from the awareness campaign Pole to Paris, which engaged non-academic audiences on climate change issues on the roads from the polar regions to Paris and through conventional and social media.*

Excerpt of the 3rd paragraph in Sect. 3:

*This positive message of a younger generation working for an act on climate was the common theme for these three videos, which also included a more simply produced video on the motivation for why the main runner and cyclist left their offices in climate research to engage with the society at large (with almost 40,000 views and a reach of nearly 150,000).*

Excerpt of the 6th paragraph in Sect. 3:

*Moreover, more than half (20 out of 37) indicated that they learned something new through Pole to Paris, signalling the potential scientists have in bridging the gap between academia and the [general] public on fundamental societal issues.*

Excerpt of the 1st paragraph in Sect. 4:

*Both academic and non-academic members of society, especially the younger ones, expressed their enthusiasm regarding the project.*

Excerpt of the 2nd paragraph in Sect. 5:

*The advancement of science might be of little significance if it is ignored by government as well as the laypeople and not suitably utilised by an educated society.*

Excerpt of the 3rd paragraph in Sect. 5:

*For scientists at the beginning of their academic career, we support the notion by Leshner (2007), Brownell et al. (2013), Rauser et al. (2017) and Nisbet (2018) that engaging in outreach activities helps shape the research questions, giving more effective tools for narrowing the widening gap between academia and the rest of society, and eventually providing a more constructive input for policy formulation on climate change.*

Breaking the filter bubbles

This paper makes an important observation about the need for scientists to engage in dialogue, especially face to face communication. These recommendations are aligned with a recent report by Matt Nisbet, for AAAS ([https://www.aaas.org/sites/default/files/s3fs-public/content_files/Scientists%2520in%2520Civic%2520Life_FINAL%2520INTERACTIVE%2520082718.pdf](https://www.aaas.org/sites/default/files/s3fs-public/content_files/Scientists%2520in%2520Civic%2520Life_FINAL%2520INTERACTIVE%2520082718.pdf)), which explains the need for scientists' engagement in civic life. Discussing this would also be valuable in the context of the role of scientists/science in society. However, I am not persuaded by the authors' assertion that their outreach efforts are the solution to climate change polarization, politicization, and the "post-truth" world. First, I think there needs to be stronger evidence of which audiences were reached in the campaign in order to make this claim. Secondly, I think these terms must be defined and explicated if they are to be used to generate recommendations for scientists. For example, what are the causes of politicization, and why do the authors think this particular outreach approach helped resolve it? Similarly, what are the causes of polarization (it is distinct from politicization), and do the authors think the campaign helped to overcome this? Why? How? Furthermore, due to the international nature of the campaign, it would be useful to understand how the effects of the outreach varied between different nations because politicization and polarization likely vary widely amongst the different nations included in the campaign.

Thank you for bringing our attention to the highly relevant work by Matt Nisbet. This is now included in the manuscript.

Excerpt of the 1st paragraph in Sect. 4:

*Engaging in two-way interaction with a range of audiences – from farmers to senators, from preschool children to retirees and from Norwegians to Bangladeshis – provided invaluable insight to our own research questions, as also highlighted by Nisbet (2018). Fortunate with these encounters, we faced questions and concerns often far from ours, which opened our eyes and ears and widened our perspectives. As reported by Nisbet (2018) and references therein, we improved our communication and listening skills and extended our professional and social network.*

Excerpt of the 2nd paragraph in Sect. 4:

*Moreover, fostering constructive public conversations about science and society can, among others, improve decision-making, promote trust and credibility in scientific findings and strengthen democratic processes (Wooden, 2006; Nisbet, 2018), ultimately counteracting politicization and polarization of science and post-factual movements, respectively.*

Excerpts of the last paragraph in Sect. 4:

*Passion united the team and contaminated our audiences, creating better dialogues in a positive feedback loop (Nisbet, 2018).*

*Most importantly, by meeting our audiences in running shoes, on a bicycle or over a beer, we connected as humans – critical to effective science engagement (Nisbet, 2018).*

Excerpt of the 3rd paragraph in Sect. 5:

*For scientists at the beginning of their academic career, we support the notion by Leshner (2007), Brownell et al. (2013), Rauser et al. (2017) and Nisbet (2018) that engaging in outreach activities helps shape the research questions, giving more effective tools for*

*narrowing the widening gap between academia and the rest of society, and eventually providing a more constructive input for policy formulation on climate change.*

The references to Nisbet (2018) is:
- Nisbet, M.: Scientists in civic life: facilitating dialogue-based communication, American Association for the Advancement of Science, https://www.aaas.org/sites/default/files/s3fs-public/content_files/Scientists%2520in%2520Civic%2520Life_FINAL%2520INTERACTIVE%2520082718.pdf, 2018.

For definitions and explications of polarization, politicization and a "post-truth" world, please see the Historical development of the climate scientist role above.

The 3^rd and 4^th paragraph in Sect. 4 discuss how we tried to overcome the polarization and politicization of climate change. Based on your suggestions, we have added some more information about how we did so. They now read:

*Consequently, we worked hard to keep our credibility as researchers (Nordhagen et al., 2014), not partnering with organizations or initiatives on either of the climate advocacy fringes, and not favouring one political party over another. Based on the feedback received, this scientific background and endeavour to remain objective allowed us to partner with organisations otherwise not within reach, like the United Nations Development Programme (UNDP) and the World Meteorological Organization. Following the definitions by Nordhagen et al. (2014), we experienced a boost in personal and public credibility, more than outweighing a loss in professional credibility from our publication record hiatus while on the road, thus overall enhancing our researcher credibility. By being open about what role we played in public, we strove to negotiate the tension between our professional and public credibilities discussed by Nordhagen et al. (2014), in which our goal of stronger climate action on governmental level to some degree was challenged by the common academic view that researchers should remain detached from public policies. We saw our role as awareness-raisers, increasing the understanding of climate science within all societal groups. Spanning the cultural differences within these groups, we tailored the message to the audiences in line with the suggestions by Somerville and Hassol (2011). These included framing climate change as a human and not only an environmental issue, focusing on the now instead of the decades ahead, leading with what we know, using a language adapted to a public discourse, being passionate, and connecting the dots between climate change and the personal experiences of the audience themselves.*

*The ten languages spoken by the highly international Pole to Paris group members helped in this way by allowing us to personally engage with a wide range of people on the roads from the polar regions to Paris. Besides, these language skills helped spread our messages even further, as suggested by the 62 % followers on Facebook speaking English, 16 % Indonesian, 6 % Norwegian, 4 % French, 3 % Spanish and 2 % German. Similarly, as suggested by Wooden (2006), the collaboration with local partner institutions (e.g., Gateway Antarctica in New Zealand, the Bjerknes Centre for Climate Research in Norway, the UK Youth Climate Coalition in the UK and Climate Generation in USA) offered experience for successful ways of science communication within each country. This collaboration also allowed us to organize what we called the Global Voices events with our partner UNDP. These were set up outside the routes of the Northern Run and Southern Cycle (Fig. 1),*

*during which youth came together to learn about climate change and how they could act upon it.*

Renaming of title
Does the title clearly reflect the contents of the paper? No. As I explained, delving into a discussion about the "post-truth" era, polarization, and politicization requires much more explication and a different kind of data than what is provided here. I suggest renaming the paper.

We appreciate you sharing your opinions and recommendations on this matter, which we believe help us strengthen the manuscript. We have therefore tried to follow your suggestions as much as possible in the updated manuscript. Here, we tell the story of being young environmental scientists having tried to actively bridge what we see as a widening gap between science and populism, building on our experiences from Pole to Paris (the two authors ran about 2450 km and 750 km of the Northern Run in addition to backing other parts of the project, including the Southern Cycle) and other environmental awareness projects that we have been involved in.

In line with this, we have done several changes to the text, as explained above and highlighted in the updated manuscript. We have also changed the manuscript title to reflect this storyline, from "The role of climate scientists in the post-factual society" to "The role of climate scientists in the post-factual society: Reflections from the awareness campaign Pole to Paris". With these changes, we hope and believe that the manuscript now tells a clearer story, in which we share our experiences to contribute to the scientific discussion on what role climate scientists should consider playing in the 21st century.

[revised manuscript text omitted]

**Slettet:** -
**Formatert:** Innrykk: Venstre: 1,27 cm, Ingen punktmerking eller nummerering
**Slettet:**
**Formatert:** Innrykk: Venstre: 1,27 cm, Ingen punktmerking eller nummerering
**Slettet:**
**Formatert:** Innrykk: Venstre: 1,27 cm, Ingen punktmerking eller nummerering
**Slettet:**
**Formatert:** Innrykk: Venstre: 1,27 cm, Ingen punktmerking eller nummerering
**Slettet:**
**Formatert:** Innrykk: Venstre: 1,27 cm, Ingen punktmerking eller nummerering
**Slettet:** 'pure scientist'
**Slettet:** 'science communicator'
**Slettet:** is completely pointless
**Slettet:** general public
**Slettet:** .
**Slettet:** ) and
**Slettet:** helps shaping
**Slettet:** general public
**Slettet:** this

[revised manuscript text omitted]

---

## Author Response (AR1)

**The role of climate scientists in the post-factual society: Reflections from the awareness campaign Pole to Paris**

Erlend M. Knudsen and Oria J. de Bolsée

**Set-up of response**

We thank the editor for her suggestions on the manuscript. With the changes explained below, we feel that the paper is strengthened compared to its first and second submissions.

In the following, we go through each comment by the editor (reproduced here in gray text for your reference) and explain our choices of changes in accordance with these. Where changes to the text in the manuscript are made, the relevant excerpt is reproduced from the .pdf manuscript to this .docx response in *italic text*, with changes written in *italic green text*.

**Response to the review by Heidi Roop**

Climate change messages and engagement

Based on the responses, the manuscript still requires more structure and clarity in regards to the analysis used, as well as more structure to make it clear to the reader how they arrived at some of the specific recommendations made at the end of the manuscript. More initiative-specific information is required to help the reader understand a) how specifically their messages differed from the climate change messages they critique, and b) how specifically they structured their engagements to support dialogue. The effort to build dialogue is a central thesis of the paper, but how this was done in the initiative is not entirely clear. How specifically were the public talks set-up to support dialogue, rather than a deficit-based interaction? Or, was the dialogue developed through those who joined in the cycling and running part of the engagement? If so, can the authors describe who it was that joined in the physical components of the efforts? How many people does that roughly represent from your efforts and how did they learn about the opportunity to participate?

Thank you for pointing out where you find clarity to be missing in the manuscript in light of the changes done to accommodate the reviews of the reviewers. We have done our best to structure the manuscript better and make it clearer.

More than how the climate change messages differed within the Pole to Paris initiative in comparison to most outreach initiatives, the way they were delivered differed more. As we write in the end of Sect. 4, we met our audience in running shoes, on a bicycle or in other informal settings, being accessible and using an appropriate language and format in our society instead of asking our audiences to adapt to our forums, formats and terminology.

We established a two-way interaction by:

- Researching about our audience prior to meeting them to formulate our message appropriately. For example, we called the teacher at the upcoming school presentation to know what the students have learned about recently and thus relate to. Similarly, we read local news ahead of open town events to make sure to relate our stories to stories of their own.
- Starting presentations by asking our audience what they already knew about the topic to make sure the information shared was relevant, well paced and detailed and not a repetition of what they already knew. While this took 5 minutes off the given time slot, we found it to be worth the time due to the improved connection with the audience and their enhanced willingness to ask questions throughout our presentations as they already had raised their voices once.

- Using a language that is accessible and appropriate. As we highlight in Sect. 4, the various professional and demographic backgrounds of the Pole to Paris team allowed us to reach a wider audience than if the team members were all male, white, scientists and only speaking English.
- Adopting an attitude that put our audience at ease and encouraged them to participate and ask questions. Essential here was making sure that the audience saw us more as humans with knowledge and passion of the environment rather than scientists from an ivory tower.
- Creating games and activity-based interactions which by nature created a more laid-back atmosphere of exchanges and questions. For instance, we would create a game starting with a discussion on solutions to personally act on climate change that participants were aware of. This alone would trigger debate among them and exchanges with us. Eventually, each participant would commit to a specific action that he or she would stick to in order to address climate change at his or her scale.

We used this methods in all our work, but two of our outreach forms stand out in establishing a two-way interaction: 1) events with local partner organisations in open spheres, where most time and focus were on individual or group interaction with the laypeople, and 2) meetings with locals along the ways from Tromsø and Christchurch to Paris – running or cycling with us for parts of the journeys or inviting us into their homes.

The latter were a mix of people, from athletes, who wanted to a stronger meaning to their training, to people of all shapes, who got inspired to leave their houses to run or cycle for a good cause. During the Northern Run, the lead runner always ran with a GPS tracker in his or her backpack, which allowed anyone to follow the journey through a live map on our website and join in the run as he or she ran through their neighbourhood. The awareness of this option was spread through conventional and social media and through our networks.

In total, we estimate that about 500 people ran and cycled with us along parts of the Northern Run and the Southern Cycle, with the majority being during the open town/city events of the latter. By nature, the largest cities with the highest potential of running followers along the Northern Run were towards its end, i.e., London, Brussels and Paris. However, due to the Paris attacks and the following lockdown in Brussels and France, the events planned in these cities, with interest from hundreds of co-runners, either had to be cancelled or strongly downscaled, i.e., without the joining runners.

1) Clearer methods and goal sections

The paper will still benefit from clearer methods and goals sections. For example, Section 2, still needs a clear statement of the intended target audience of the initiative. These goals can then be positioned against the data collected to reflect on the extent to which the stated goals were achieved.

Based on your suggestions, we have rewritten and split the 2nd paragraph in Sect. 2 in two. These read:

*The Pole to Paris project focused on reshaping the way scientists engage with the public on climate change issues. The nature of the problem – being a long-term process on a planetary scale – makes it difficult for individuals to grasp and engage with. In an attempt to remove this abstractness, we, as scientists, decided to hit the road in order to share climate science knowledge with people on the ground as well as collect their stories of experienced changes to share them through our platforms. This allowed us to target audiences along the way not normally reached by scientific messages, meeting them face-to-face. Instead of inviting them to our universities, using a scientific jargon and sharing scientific information behind paywalls, we met them on their terms – in their home forums, using a familiar language and connecting through accessible formats.*

*To reach this audience, two journeys from the poles were mapped out: the 10,000-km long bicycle ride – the Southern Cycle – from Christchurch (New Zealand) and the 3,000-km long run – the Northern Run – from Tromsø (Norway), both finishing in Paris during COP 21 (Fig. 1). These journeys were led by two climate scientists, who left Christchurch and Tromsø shortly after completing their PhDs in Antarctic and Arctic climate change, respectively. 7.5 and 4 months later, respectively, they reached Paris, carrying flags from the melting polar regions and stories from people met along the way. The two were supported by the eight other Pole to Paris team members, whose backgrounds ranged from environmental and political science to web and product design. While all members actively contributed to Pole to Paris by various means from their locations around the world, five of them also joined the main cyclist and runner for part of the journeys. Of the ten team members, only the main cyclist and runner were working full-time on the project (i.e., without getting paid), while the others had studies or jobs to balance simultaneously. Whereas we were all in our 20s, the four female and six male team members represented eight different countries.*

2) Reflection on the Pole to Paris experience

As noted by the reviewers, the paper needs to include "discussion reflecting more on the experience" of the Pole to Paris effort. Specifically, one reviewer noted "The authors could strengthen this paper by spending more time reflecting on and discussing the content of their climate messages and exploring where those messages sit on a spectrum from objective to advocate." There are several areas where this can still be explicitly included in the manuscript to respond to both reviewers' concerns and suggestions.

Based on your suggestion, we have added a new 4th paragraph in Sect. 3, expanded the 3rd paragraph in Sect. 4 into two paragraphs and adopted the beginning of the 5th paragraph accordingly. These excerpts now read:

*As environmental scientists, who had tried to engage the people around us on climate change and biodiversity loss prior to Pole to Paris, the authors find the popularity of the climate action videos encouraging. However, this also questions our objectivity as scientists. Through the videos, we advocated for personal and societal action on climate change, as we did in media and our presentations. Hence, we moved beyond our core*

*scientific base and took on roles as the 'science communicator' and 'the honest broker of policy alternatives,' as defined by Rapley and De Meyer (2014). We found this necessary due to the nature of the problem – often seen as something far away in space or time. By sharing stories of climate change our audience could connect to, we made the problem more visible and graspable - to something right here, right now. This established connection also raised a willingness to do something about the problem, which we advocated for through the reduction of personal greenhouse gas emissions, through the investment power of consumers and companies and through bringing the problem into light among family, friends and colleagues. Had we only communicated the threat of climate change without making it relevant and suggesting ways the listener could address the problem, we would have created a maladaptive response (e.g., denial) among our audience, according to Witte (1992).*

*By being open about the role we played in public, we strove to negotiate the tension between our professional and public credibilities discussed by Nordhagen et al. (2014), in which our goal of stronger climate action on a governmental level was challenged to some degree by the common academic view that researchers should remain detached from public policies. However, as Kotcher et al. (2017) point out, this notion is not supported by empirical evidence. On the contrary, in line with their results, we experienced no direct harm to our public credibility or to that of the scientific community.*

*Considering the time span over which the analysed videos were posted, the later videos were generally more popular. This points to the increasing reach of Pole to Paris as the awareness project gained traction with kilometres covered, events held along the way, and mentions in the media. Even when the project reduced its activity after COP 21, the influence was still there, as exemplified by reaches of more than 100,000 on the less frequent Facebook posts in early 2016.*

The new reference is:

- Kotcher, J. E., Myers, T. A., Vraga, E. K., Stenhouse, N., and Maibach, E. W.: Does engagement in advocacy hurt the credibility of scientists? Results from a randomized national survey experiment, Environ. Commun., 11, 415–429, doi:10.1080/17524032.2016.1275736, 2017.

2a) Sharing of climate change stories

In section 2, the authors lead with the goal of the project being to share climate knowledge with people on the ground and to "...collect their stories of experienced changes and share them through our platforms". This aspect of the project is not addressed in the manuscript. What were those stories? Are these some of the videos discussed in the analytics? How did the stories and message differ or resonate with the audiences vs. the videos featuring the scientists at the heart of the effort?

Based on your suggestion, we have added a new 5th paragraph in Sect. 4, which reads:

*For establishing personal connections to climate change among our audiences, we found that sharing personal experiences of climate change from people we met along the way was especially successful. As scientists, we are used to speak in terms like 2°C, 450 ppm and 50 cm, but most people cannot relate to these numbers. Rather, they relate to stories of people like them whose livelihoods are threatened by climate change. Consequently, we listened to stories like those of a Sami, who might not be able to pass the reindeer herding tradition on to her children due to the warming winters; of a Bangladeshi,*

*who might become a climate refugee due to the rising sea; and of a Londoner, who might be protected from the worst consequences in the metropolis but chooses to write about global environmental issues and work with organisations to find solutions. We shared these stories and others from the road through conventional and social media and in presentations on the way to Paris, at a press conference and at the conference centre there and in a documentary and a TEDx talk since. Based on the video analysis alone, it is difficult to say that these messages were most popular, partly because we did not feature them all in videos and partly because they were both more and less popular than the videos featuring the scientists at the heart of the effort. However, based on interaction with journalists and our audiences, we have strong reasons to believe that these personal stories strongly helped in making the climate science relatable.*

2b) Dialogues and two-way conversations

The authors refer in several instances to "dialogue" and "two-way conversation", but don't provide any specifics about how they took a traditional 'deficit' approach of giving talks in "schools and universities" and made them dialogical in nature and how they reached non-academic audiences in these engagements. Given this is a central argument of the paper, as a reviewer noted, more attention needs to be paid to this, as it seems these are the key fora in which the authors undertook 'dialogue' and is the supporting evidence for their call that other climate scientists should do more of this work. As noted by a reviewer, the authors still need to consider the addition of Public Engagement with Science literature around dialogue and two-way interaction, given the numerous references to this approach.

Based on your suggestion, we have added a new 6th paragraph in Sect. 2, which reads:

*Our approach thus differed from the information deficit model, as outlined by Bucchi (2008). In this model, the public is considered passive and ignorant. Its hostility to science can be counteracted by appropriate injection of science communication, which is provided by experts (i.e., scientists) through a linear, one-way process to non-experts (the public) (Bucchi, 2008). However, this top-down approach is no longer appropriate for our current society, where science communication is addressing a wider agenda (Bucchi, 2008). Instead, the need and right of the public to participate in the scientific discussion has led to dialogue and knowledge models through which the involvement of lay people have enhanced the competencies of scientists and specialists (Callon, 1999; Trench, 2006). We found the latter models to be highly rewarding, as we learned a lot from the dialogues ourselves in addition to being better understood as communicators of scientific information.*

Based on your suggestion, we also added a sentence to the 2nd paragraph in Sect. 4. This excerpt now reads:

*Meeting people where they are, in their own communities, communicating with them in their own terms, constantly trying to adapt our language to our audience, undeniably contributed to this. We connected through dialogue. Considering the politicized division of the media themselves (e.g., Brüggemann and Engesser, 2017), this positive experience of direct engagement supports the suggestion by Gauchat et al. (2017) that science participation and outreach could rebuild the credibility among communities most critical of scientists.*

The new references are:

- Bucchi, M.: Of deficits, deviations and dialogues: Theories of public communication of science In Handbook of public communication of science and technology, 71–90, Routledge, Abingdon, UK, ISBN:978-0-415-38617-3, 2008.
- Callon, M.: The role of lay people in the production and dissemination of scientific knowledge, Sci. Technol. Soc., 4, 81–94, doi:10.1177/097172189900400106, 1999.
- Trench, B.: Science communication and citizen science: How dead is the deficit model?, Scientific Culture and Global Citizenship, Ninth International Conference on Public Communication of Science and Technology (PCST-9), Seoul, Korea, May 17-19, 2006, 2006.

2c) Targeting the climate messages + Framing of climate messages

More context is needed to make it clear how specific parts of the Pole to Paris messaging was made 'relevant', as this is a key suggestion at the end of the article. What specific messages were relevant to your broad audience and do the analytics show this? What were the climate messages you tried and how did you modify messages for the different audiences you described engaged with across international borders, values, and languages? Additionally, the descriptions of the videos don't make it clear how the authors "highlighted the opportunities and inspiration of acting on climate now rather than later". Specifics about how they themselves deployed specific messages and developed their best practices are still needed in order to clearly demonstrate exactly what messaging was used, so that readers can see how their suggestions at the end of the article fit into the context of the work the authors carried out.

Based on your suggestion, we have rewritten and split the previous 3rd paragraph in Sect. 2 in three. These read:

*The public were invited to get behind the Southern Cycle and Northern Run journeys and actively become engaged in the climate dialogue in real time. This was partly done online through social media, partly at the events through open accessibility and partly on the roads themselves through planned and improvised meetings. To some extent, the latter happened because of GPS tracking on our website (Fig. 2), which allowed for other cyclists and runners to join us for part of the distances, providing an accessible and informal platform for face-to-face dialogues. The adventure component also helped to attract media attention, giving the project a platform to communicate the facts about climate change and the importance of COP 21 to the wider audience by engaging them in the journeys. Crucially, along the way, we held talks in schools, universities and many other public venues. To make our climate messages engaging, we called the teacher and read the local news ahead of the presentations to identify topics our audiences could relate to. The former also allowed for the students to be prepared for our presentations, following us online and learning about relevant material prior to our visit.*

*The ironic beauty of the climate change problem is that is encompasses the whole society, from health and food to tourism, migration and the economic system. Hence, we could always bring our climate messages into a familiar context for our audiences and thus stimulate their feedback. This was also helped by often starting presentations asking the audience what they already knew about the topic in a humane and positive attitude that set everyone at ease. Similarly, we created games and activity-based interactions, especially for our youngest audiences, which brought the large-scale climate problem down to his or her scale. Even though this took time from our given time slots, we found this to better adopt the pace and detail level of our climate messages while also lowering the threshold for questions*

*and comments from the audience. Altogether, this created a true dialogue, in which we openly engaged the public to hear their perspectives and concerns about climate change before respectively responding to them, as suggested by Leshner (2003).*

*We collaborated with our partners to create events, and we shared stories from the road through conventional and social media (Fig. 2). This provided a unique opportunity to interact with members of society not usually reached by the scientific discourse. In line with O'Neill and Nicholson-Cole (2009) and Stoknes (2015), we highlighted the opportunities and inspiration of acting on climate change now rather than later. For example, from an economical viewpoint, strong, early climate action considerably outweighs its costs (Stern, 2007). Similarly, from a job market perspective, more jobs are added in the energy industry within renewables than are lost in fossil fuels (Fankhaeser et al., 2008). We still communicated the dangers associated with ongoing and expected consequences of climate change, but in terms of relevant and experienced changes rather than fear rising from their cognitive dissonance following Extended Parallel Processing Model theory (Witte, 1992). This theory suggests that such messaging promotes a protection motivation and thus a willingness to change in accordance with the message for the recipient, in contrast to a defensive motivation and thus a reluctance to change (e.g., denial).*

The new references are:

- Fankhaeser, S., Sehlleier, F., and Stern, N: Climate change, innovation and jobs, Clim. Policy, 8, 421–429, doi:10.3763/cpol.2008.0513, 2008.
- Stern, N.: The economics of climate change, Cambridge University Press, Cambridge, UK, ISBN:978-0521700801, 2017.

3) Number of audiences reached + Transparency of statistical analysis

One reviewer noted that the "the descriptions of the audiences reached through the effort must be more precise" in the analytics section. This still needs to be addressed. Perhaps these numbers are better suited to a table? Further, the analysis of the social media analytics does not appear statistical in nature. If statistical analyses were used, how specifically were the data treated? Also, more information about the post-project evaluation, and some of the approaches used, including who and how their social media campaigns were paid for, and on what platforms, would create more transparency and help readers to understand the nature of the analytics and their approach to data collection. All of this methodological information could be more explicitly stated at the top of the "Direct successes" section. The authors also conducted a post-project online survey and might consider sharing the survey questions in an appendix.

Based on the feedback from the reviewers, we expanded the information about the number of audiences reached from a sentence to an own paragraph at the end of Sect. 2. We are sorry to hear that this still is not sufficient.

The idea of collecting all numbers in a table is attractive. We would have supported this idea if all numbers were of comparable formats. Unfortunately, because of the very different representations of these numbers (e.g., social media channels provide number of people reached for each story while conventional media houses only provide overall number of printed newspapers or TV station availability), we found such a table to be more confusing to the reader than clarifying.

However, in an effort to make our data analysis as transparent as possible, we have added a new Fig. 2 showcasing parts of our website and social media channels, as well as spreadsheets of the publicly available media coverage (discussed in Sect. 2; Table A1) and statistics of Facebook videos (discussed in Sect. 3; Tables A2 and A3) and graphics of the social media survey (discussed in Sect. 3; Fig. A1). This will hopefully clarify the data treatment for the Facebook videos statistics and social media survey, which came from Facebook and SurveyMonkey, respectively.

Based on your suggestions, we have added the new Figs. 2 and A1 and Tables A1-A3 and references to them in relevant excerpts in the 4th, 7th and last paragraphs in Sect. 2 and 1st and 7th paragraphs in Sect. 3. These now read:

*This was partly done online through social media, partly at the events through open accessibility and partly on the roads themselves through planned and improvised meetings. To some extent, the latter happened because of GPS tracking on our website (Fig. 2), which allowed for other cyclists and runners to join us for part of the distances, providing an informal platform for face-to-face dialogues.*

*We collaborated with our partners to create events, and we shared stories from the road through conventional and social media (Fig. 2).*

*A conservative estimation is that more than one million people in 45 countries were reached through conventional and social media, which included about 250 media outlets (Table A1) and almost 500,000 and 250,000 reached per Facebook post and Twitter tweet, respectively.*

*Data for this analysis was fetched through the export function that Facebook offers for administered pages. In addition to information about the date videos were published, links to them and their titles, this function provides information about unique and total views, organic and paid views, and views after 3 seconds, at least 30 seconds (or to their end if that came first) and at 95 % of the video length (including viewers that skipped to this point). We subjectively categorized the videos by topic and main country(ies). Of the 42 total videos, we focused the analysis on the 32 in the most active period from June to December 2015. Detailed data on these can be found in Tables A2 and A3 in the Appendix.*

*The survey was set up through the online survey platform SurveyMonkey and asked the anonymous respondents a range of questions (Fig. A1).*

4) Shaping scientific questions

In several places the authors argue this type of engagement work "helps to further develop scientific questions", but the authors do not elaborate on that element of their own work in this manuscript or make it clear that was an intended goal or approach of the initiative. If this is true for the authors, please elaborate and reference a greater breadth of literature (e.g. from the field of knowledge co-production/actionable science). If not, these references might be removed as they appear tangential to the central arguments/experiences of the authors and this work.

Unfortunately, we were not able to find the statement that engagement work like Pole to Paris "helps to further develop scientific questions". Hence, while being positive in changing the manuscript in line with your suggestions, we find it difficult to address this comment.

In the first paragraph in Sect. 4, we write "Engaging in two-way interaction with a range of audiences [...] provided invaluable insight to our own research questions, as also highlighted

by Nisbet (2018).” What we meant by this is that questions and concerns of non-academic members of society often strongly differ from those within academia, as we also write in the sentence following: “Fortunate with these encounters, we faced questions and concerns often far from ours, which opened our eyes and ears and widened our perspectives.”

In the last paragraph in Sect. 4, we write “While we strongly acknowledge the need for publishing research papers to further develop scientific questions, we emphasize that the findings thereof are incomplete if not shared with the society at large.” What we mean by this statement is that publishing research papers should still be part of the scientific role as this is important for bringing scientific understanding further. However, as we see it, the parts of society that gets to thrive on this scientific understanding is limited to academia unless the results of the published papers are shared with the rest of society through accessible means and terms.

5) Reference to Witte (1992)

The reference to Witte 1992, as suggested by needs to be elaborated on. As currently incorporated, it is unclear how the authors see this theory as part of their approach or work.

Based on your suggestion, we have included a better explanation on how we made use of the Extended Parallel Processing Model theory by Witte (1992). Hence, the relevant excerpts in the 7th paragraph in Sect. 2 and the 4th paragraph in Sect. 3 now read:

*We still communicated the dangers associated with ongoing and expected consequences of climate change, but in terms of relevant and experienced changes rather than fear rising from their cognitive dissonance following Extended Parallel Processing Model theory (Witte, 1992). This theory suggests that such messaging promotes a protection motivation and thus a willingness to change in accordance with the message for the recipient, in contrast to a defensive motivation and thus a reluctance to change (e.g., denial).*

*Had we only communicated the threat of climate change without making it relevant and suggesting ways the listener could address the problem, we would have created a maladaptive response (e.g., denial) among our audience, according to Witte (1992).*

6) Reframing of the abstract

The abstract still needs to be adapted to support the main arguments of the paper. Both reviewers suggested a reframing of the abstract, but no edits have been made. Given the significant changes to the manuscript, changes to the abstract are warranted.

Thank you for pointing the missing changes to the abstract following the update of the manuscript.

Based on your suggestions, the abstract now reads:

*The politicization of and societal debate on climate change science have increased over the last decades. Here, the authors argue that the role of climate scientists in our society needs to adapt in accordance with this development. We share our experiences from the awareness campaign Pole to Paris, which engaged non-academic audiences on climate change issues on the roads from the polar regions to Paris and through conventional and social media. By running and cycling across a third of the globe, the scientists behind the initiative established connections on the audiences’ terms. Propitiously for other outreach efforts, the exertions were not in themselves the most attractive; among our social media*

*followers, the messages of climate change science and action were more favourable, as measured by video statistics and a follower survey. Communicating climate action in itself challenges our positions as scientists, and we here discuss the impact such messages have on our credibility as researchers. Based on these reflections, as well as those from other science communication initiatives, we suggest a way forward for climate scientists in the post-factual society, who should be better trained in interaction with non-academic audiences and pseudoscepticism.*

7) Renaming of the title

Both reviewers suggested a new title. The framing around the 'post-factual' society, while more thorough in this version of the manuscript, still does not seem to be the central thesis of the paper. The last section of the paper "an adapted scientist" might be something the authors consider incorporating into the title.

Based on your suggestion, we have changed the title from "The role of climate scientists in the post-factual society: Reflections from the awareness campaign Pole to Paris" to "Adapted climate scientists to a post-factual society: Reflections from the awareness campaign Pole to Paris."

Compared to the original version of the manuscript, including its title "The role of climate scientists in the post-factual society", we have elaborated much more on what is meant by a post-factual society. This is seen in Sect. 1 from a general perspective and Sect. 4 from a Pole to Paris perspective. With the inclusion of the subtitle "Reflections from the awareness campaign Pole to Paris", we specify our viewpoint to be from mostly one awareness campaign. Moreover, by changing "The role of climate scientists" to "Adapted climate scientists", we are – as we see it – moving from a possible interpretation of that all climate scientists need to be this way to that climate scientists could be this way. Finally, by replacing "the post-factual society" by "a post-factual society", we clarify that we mean that not our whole society is post-factual; rather, portions of our society is led by post-factual movements, which value emotions and beliefs over scientific facts and elect leaders thereafter. The latter, with their negligence and denial of climate science, is the key reason why we initiated Pole to Paris in the first place and finally why we wanted to write about our experiences with it in a manuscript.

We hope that we have better clarified the reasons for the title and made sufficient adjustments to it to reflect the messages in the manuscript.

8) Suggestions for an adapted scientist

Along those lines, the suggestions for the adapted scientist seem out of context at the end of the paper. These suggestions, and more explicit examples of how the Pole to Paris effort carried out these suggestions could be better integrated into the manuscript so it is clear how what was done in the initiative led to, or modeled, these suggestions. Further, the authors note in passing that the analysis of the social media analytics occurred well- after the experience, when some data from their social platforms were no longer available. Do the authors have any insights for others regarding data collection, audience segmentation, or documentation that they think would help 'the adapted scientist'? This is noted on page 8 and might be worthwhile incorporating more thoughtfully in their recommendations at the end of the manuscript.

We thank you for stressing the importance of the link between the final suggestions for the adapted scientist and the text leading up to it. Based on your suggestion, we have gone through the manuscript once more to make sure our suggestions for the adapted scientists were rooted in the text. Were we felt they were not, we have rewritten or added text in the relevant paragraphs, as indicated by the green text below.

The importance of relevance is discussed in the following excerpts:

- Abstract:

[revised manuscript text omitted]

To clarify that the suggestions listed at the start of Sect. 5 result from the text before, we have rewritten the sentence leading up to the list and slightly modified the list itself. It now reads:

*Based on the experiences outlined above, we identified some key components for successful science communication with non-academic audiences:*

- *Relevance*

  *Make sure your message is relevant to your audience and engage with them in familiar setting, with a familiar format and through a familiar language.*

- *Listening*

  *Let the audience ask questions and describe their understanding in their own words.*

- *Positivity*

  *Smile and try to focus on the possibilities rather than the doomsday scenarios.*

- *Perseverance*

  *Learn by doing and keep doing it; all experiences are valuable.*

- *Passion*

  *For communicating science, knowledge of the topic is essential, but passion is the key for the audience to absorb it.*

9) Proofreading

As noted by the reviewers, a detailed proof-read is required to tighten up the manuscript.

Based on your suggestions, we have gotten a native English speaker to proofread the manuscript. Additionally, we have both proofread the manuscript ourselves. These minor edits are too many to be reproduced here but can be found in the tracked changes version of the manuscript.

[revised manuscript text omitted]

**formaterte:** Ikke Hevet/ Senket

**formaterte:** Ikke Hevet/ Senket

**slettet:** , ahead and

**slettet:** ,

**slettet:** them through our platforms. Two

**slettet:** . They

**slettet:** the

**slettet:** Whereas we

**slettet:** represented

The public were invited to join the Southern Cycle and Northern Run journeys and actively engage in the climate dialogue in real time. This was partly done online through social media, partly at the events through open accessibility and partly on the roads themselves through planned and improvised meetings. To some extent, the latter happened because of GPS tracking on our website (Fig. 2), which allowed for other cyclists and runners to join us for part of the distances, providing an accessible and informal platform for face-to-face dialogues. The adventure component also helped to attract media attention, giving the project a platform to communicate the facts about climate change and the importance of COP 21 to the wider audience by engaging them in the journeys. Crucially, along the way, we held talks in schools, universities and many other public venues. To make our climate messages engaging, we called the teacher and read the local news ahead of the presentations to identify topics our audiences could relate to. The former also allowed for the students to be prepared for our presentations, following us online and learning about relevant material prior to our visit.

**slettet:** get behind

**slettet:** become engaged

**slettet:** and were joined by other cyclists and runners for part of the distances. This created a two-way communication

[revised manuscript text omitted]

**slettet:** that

**slettet:** stay over

**slettet:** through

**slettet:** that

**slettet:** by Pole to Paris

**slettet:** were

**slettet:** in place

The ten languages spoken by the highly international Pole to Paris group members helped in this way by allowing us to personally engage with a wide range of people on the roads from the polar regions to Paris. Besides, these language skills helped spread our messages even further, as suggested by the 62 % followers on Facebook speaking English, 16 % Indonesian, 6 % Norwegian, 4 % French, 3 % Spanish and 2 % German. Similarly, as suggested by Wooden (2006), the collaboration with local partner institutions (e.g., Gateway Antarctica in New Zealand, the Bjerknes Centre for Climate Research in Norway, the UK Youth Climate Coalition in the UK and Climate Generation in USA) offered experience for successful ways of science communication within each country. This collaboration also allowed us to organize what we called the Global Voices events with our partner UNDP. These were set up outside the routes of the Northern Run and Southern Cycle (Fig. 1), during which youth came together to learn about climate change and how they could act upon it.

The experiences from Pole to Paris were, however, not unique. Other initiatives have been launched over the last few years to increase climate change awareness and train scientists in more effective science communication. We were some of the 1.07 million people who took part in the March for Science on April 22, 2017. The series of rallies and marches defending the vital role science plays in our everyday lives was a direct result of the opposing direction on science policy taken by the current administration in the White House compared with its predecessor. However, March for Science has also been criticized, as it runs the risk of creating a false picture of scientists being more driven by ideology than evidence (Nature supports the March for Science, 2017).

**slettet:** globally to take

Furthermore, the authors have been involved in other more or less politically charged outreach projects. For instance, Climate Communication Cologne is an effort launched at the University of Cologne whose main objective is to facilitate science communication to a wide non-academic audience. This takes place in various forms, such as workshops, stand-up comedy or videos, and in various arenas, from schools and universities to pubs and online communities. Another example is Will You Hear Us, a documentary on the tradition of caged birds in Indonesia, which has become unsustainable due to the ever-increasing demand for wild songbirds and poses a huge threat on biodiversity. Both authors are currently also writing comic books on climate change adaptation and mitigation and on biodiversity loss for high-school and elementary school students, respectively.

Common for all these initiatives is the eagerness to communicate science in ways that engage the layperson. To help us – and the reader of this manuscript – to learn from our efforts, we ideally would have set up a more standardized feedback scheme for our audiences during the active period of Pole to Paris. The feedback we did receive – in personal conversations and in online commentary forums – were most likely anomalously positive and negative, respectively. We could surely also have benefitted from more planning and training before undertaking these journeys, but this might have compromised the journeys themselves. Being the only two full-time-engaged team members , the two climate scientists of Pole to Paris – the lead cyclist and runner – had just completed their PhDs before taking on the journeys, while the other eight in the team had full time commitments to studies or employers to balance, which did not provide much room for further planning. This, along with the widely varying time zones the team members were based in and frequent lack of internet accessibility out on the Southern Cycle and Northern Run, meant that team meetings were less regular than what would have been ideal for making sure we were all pulling in the same direction.

**slettet:** fora

**slettet:** fully “working” (i.e., without getting paid) on the project

**slettet:** often

Passion united the team and was contagious amongst our various audiences, creating better dialogues in a positive feedback loop (Nisbet, 2018). We cycled and ran out with rough plans and adapted along the way as engagement created opportunities (e.g., the Global Voices events and United Nations program partnerships) or disasters imposed limitations (e.g., the Nepal earthquake and Paris terror attacks). Similarly, even though we had scientific and professional communication training to start with, we learned a lot by doing. Most importantly, by meeting our audiences in running shoes, on a bicycle or in other informal settings, we connected as humans, which is critical for effective science engagement (Nisbet, 2018). While we strongly acknowledge the need for publishing research papers to further develop scientific questions, we emphasize that the findings thereof are incomplete if not shared with the society at large.

**slettet:** contaminated

**slettet:** communicational

**slettet:** over a beer

**5 An adapted scientist**

Based on the experiences outlined above, we identified some key components for successful science communication with non-academic audiences:

**slettet:** these

- Relevance
  Make sure your message is relevant to your audience and engage with them in familiar setting, with a familiar format and through a familiar language.
- Listening
  Let the audience ask questions and describe their understanding in their own words.
- Positivity
  Smile and try to focus on the possibilities rather than doomsday scenarios.

**Formatert:** Punktmerket + Nivå: 1 + Justert ved: 0,63 cm + Innrykk ved: 1,27 cm

**slettet:** ¶

**slettet:** ¶

**slettet:** put forward

**slettet:** ¶

**slettet:** the

- Perseverance

  Learn by doing and keep doing it; all experiences are valuable.

- Passion

  For communicating science, knowledge of the topic is essential, but passion is the key for the audience to absorb it.

In our current society, we argue that the role of the 'pure scientist' (as defined by Rapley and De Meyer, 2014) is outdated and the need for the 'science communicator' and 'the honest broker of policy alternatives' (as outlined by Pielke Jr., 2007) is rising. The advancement of science might be of little significance if it is ignored by governments as well as laypeople and not suitably utilised by an educated society. Publishing an academic paper is unfinished business. As Barnosky et al. (2014) argue, it is only the beginning if our aim is to help society solve problems. However, current training of becoming scientists does not fulfil society's current need for clear science communication and policy engagement (Leshner, 2007). Thus, similar to Figueres et al. (2017), we argue that more emphasis should be placed on communication and media, policymaker and pseudoscepticism interaction training, and less on the published record.

For scientists at the beginning of their academic career, we support the notion by Leshner (2007), Brownell et al. (2013), Rauser et al. (2017) and Nisbet (2018) that engagement in outreach activities helps shape the research questions, giving more effective tools for narrowing the widening gap between academia and the rest of society, and eventually providing a more constructive input for policy formulation on climate change. As we see it, this will act to reduce politicization and polarization of climate change, while also depressing the breeding ground for post-factual movements. Within academia, outreach training gives us better tools in teaching, mentoring of younger students and taking part in scientific discussions, as well as contributing to better written research proposals and journal publications (Stiller-Reeve et al., 2016, and references therein).

Whether we like it or not, climate science and scientists have become part of the daily political and media discourse. Now it is up to us to adapt and play our new role objectively while keeping our credibility (as discussed by Nordhagen et al., 2014). According to Rapley and De Meyer (2014), this has the potential to remove climate science from the direct firing line to leave the authority, responsibility and accountability for decisions transparently with the policymakers and the public. When done carefully, we have the potential, regardless of the audience's political preferences, to provide trustworthy information to the climate change discourse (Leshner, 2003; MacInnis et al., 2015; Hamilton, 2016). To prepare us for such a "wicked" problem (as defined by Lorenzoni et al., 2007), we argue that communication training with actors beyond academia is indispensable.

**Author contributions**

EMK led the design and writing of the manuscript and carried out the statistical analysis. OJdB helped with writing and designed the social media survey.

**slettet:** ¶

**slettet:** ¶ Knowledge of the topic is the door to

**slettet:** for it

**slettet:** of

**slettet:** government

**slettet:** the

**slettet:** is

**slettet:** fulfilling the

**slettet:** of society

**slettet:** Thus,

**slettet:** put

**slettet:** engaging

**slettet:** help

**slettet:** predilection

[revised manuscript text omitted]

**formaterte:** Standardskrift for avsnitt, Understreking, Skriftfarge: Egendefinert farge(RGB(17;85;204))

**Feltkode endret**

**slettet:** s1058

**Feltkode endret**

**formaterte:** Standardskrift for avsnitt, Understreking, Skriftfarge: Egendefinert farge(RGB(17;85;204))

**Feltkode endret**

**formaterte:** Standardskrift for avsnitt, Understreking, Skriftfarge: Egendefinert farge(RGB(17;85;204))

Ungar, S.: The rise and (relative) decline of global warming as a social problem, Sociol. Q., 33, 483–501, doi:10.1111/j.1533-8525.1992.tb00139.x, 2016.

Vraga, E., Myers, T., Kotcher, J., Beall, L., and Maibach, E.: Scientific risk communication about controversial issues influences public perceptions of scientists' political orientations and credibility, Roy. Soc. Open Sci., 5, 170505, doi:10.1098/rsos.170505, 2018.

Witte, K.: Putting the fear back into fear appeals: The extended parallel process model, Commun. Monogr., 59, 329–349, doi:10.1080/03637759209376276, 1992.

Wooden, R.: The principles of public engagement: at the nexus of science, public policy influence, and citizen education, Soc. Res., 73, 1057–1063, 2006.

Zhang, B., Mildenberger, M., Howe, P. D., Marlon, J., Rosenthal, S. A., and Leiserowitz, A.: Quota sampling using Facebook advertisements, Political Science Research and Methods, 1–7, doi:10.1017/psrm.2018.49, 2018.

Zhao, X., Rolfe-Redding, J., and Kotcher, J. E., Partisan differences in the relationship between newspaper coverage and concern over global warming, Public Underst. Sci., 25, 543–559, doi:10.1177/0963662514558992, 2016.

Zürn, M.: The politicization of world politics and its effects: Eight propositions, Eur. Polit. Sci. Rev., 6, 47–71, doi:10.1017/S1755773912000276, 2014.

[Figure]

**Figure 1: Map of the two Pole to Paris journeys: the Northern Run (blue trajectory) and the Southern Cycle (red trajectory). Also included are the Global Voices events organized in collaboration with the partner United Nations Development Programme (green**

**slettet:** ); as well as

**slettet:** partners

dots).

[Figure]

Figure 2: Screenshots of Pole to Paris main website (upper left) and its web page about the Northern Run showing the interactive map with GPS tracking (upper right), Facebook (lower left), Twitter (lower centre) and Instagram (lower right) channels. Top row screenshots are from October 13, 2015, and bottom row from August 15, 2015, thus explaining the lower number of followers compared to the numbers presented in Sect. 3 from December 2015.

[Figure]

Figure 3: Percentages of total Facebook video views after three seconds (*>3 s*), at 30 seconds (or to the end, whichever came first; *≥30 s*), and at 95 % of the video length (including people that skipped to this point; *95 %*) for (a) organic (i.e., not paid; blue columns)

[Figure]

slettet:

slettet: :

**and paid (red columns) views and (b) videos on climate action (blue columns), climate change (red columns) and the journeys themselves (green columns). Numbers above the columns in (a) and (b) represent total and average views, respectively.**

---

## Editor Decision (ED1)

Editor Response: **The role of climate scientists in the post-factual society (gc-2018-16)**

Thank you to the authors for thoughtfully working to incorporate the extensive feedback from the two reviewers. The revised manuscript has evolved to be stronger in content and includes literature and context to support some of the key arguments the authors are trying to make. This paper offers an excellent case study in climate scientist-led engagement efforts and creative ways that the scientific community can engage in public discourse about climate change and climate policy. While the authors have clearly worked to incorporate some the reviewer suggestions, there is still work to do to address some of the structural and data transparency issues noted by both reviewers.

Specifically, both reviewers pointed to the need for more context and discussion of the authors' experiences and a clearer outline of the methods and assumptions used in the data analysis. Based on the responses, the manuscript still requires more structure and clarity in regards to the analysis used, as well as more structure to make it clear to the reader how they arrived at some of the specific recommendations made at the end of the manuscript. More initiative-specific information is required to help the reader understand a) how specifically their messages differed from the climate change messages they critique, and b) how specifically they structured their engagements to support dialogue. The effort to build dialogue is a central thesis of the paper, but how this was done in the initiative is not entirely clear. How specifically were the public talks set-up to support dialogue, rather than a deficit-based interaction? Or, was the dialogue developed through those who joined in the cycling and running part of the engagement? If so, can the authors describe who it was that joined in the physical components of the efforts? How many people does that roughly represent from your efforts and how did they learn about the opportunity to participate?

Some more specific suggestions, echoing those already made by the reviewers, include:

1) The paper will still benefit from clearer methods and goals sections. For example, Section 2, still needs a clear statement of the intended target audience of the initiative. These goals can then be positioned against the data collected to reflect on the extent to which the stated goals were achieved.

2) As noted by the reviewers, the paper needs to include "discussion reflecting more on the experience" of the Pole to Paris effort. Specifically, one reviewer noted "The authors could strengthen this paper by spending more time reflecting on and discussing the content of their climate messages and exploring where those messages sit on a spectrum from objective to advocate." There are several areas where this can still be explicitly included in the manuscript to respond to both reviewers' concerns and suggestions:
   a. In section 2, the authors lead with the goal of the project being to share climate knowledge with people on the ground and to "…collect their stories of experienced changes and share them through our platforms". This aspect of the project is not addressed in the manuscript. What were those stories? Are these some of the videos discussed in the analytics? How did the stories and message differ or resonate with the audiences vs. the videos featuring the scientists at the heart of the effort?

b. The authors refer in several instances to "dialogue" and "two-way conversation", but don't provide any specifics about how they took a traditional 'deficit' approach of giving talks in "schools and universities" and made them dialogical in nature and how they reached non-academic audiences in these engagements. Given this is a central argument of the paper, as a reviewer noted, more attention needs to be paid to this, as it seems these are the key fora in which the authors undertook 'dialogue' and is the supporting evidence for their call that other climate scientists should do more of this work. As noted by a reviewer, the authors still need to consider the addition of Public Engagement with Science literature around dialogue and two-way interaction, given the numerous references to this approach.

c. More context is needed to make it clear how specific parts of the Pole to Paris messaging was made 'relevant', as this is a key suggestion at the end of the article. What specific messages were relevant to your broad audience and do the analytics show this? What were the climate messages you tried and how did you modify messages for the different audiences you described engaged with across international borders, values, and languages? Additionally, the descriptions of the videos don't make it clear how the authors "highlighted the opportunities and inspiration of acting on climate now rather than later". Specifics about how they themselves deployed specific messages and developed their best practices are still needed in order to clearly demonstrate exactly what messaging was used, so that readers can see how their suggestions at the end of the article fit into the context of the work the authors carried out.

3) One reviewer noted that the "the descriptions of the audiences reached through the effort must be more precise" in the analytics section. This still needs to be addressed. Perhaps these numbers are better suited to a table? Further, the analysis of the social media analytics does not appear statistical in nature. If statistical analyses were used, how specifically were the data treated? Also, more information about the post-project evaluation, and some of the approaches used, including who and how their social media campaigns were paid for, and on what platforms, would create more transparency and help readers to understand the nature of the analytics and their approach to data collection. All of this methodological information could be more explicitly stated at the top of the "Direct successes" section. The authors also conducted a post-project online survey and might consider sharing the survey questions in an appendix.

4) In several places the authors argue this type of engagement work "helps to further develop scientific questions", but the authors do not elaborate on that element of their own work in this manuscript or make it clear that was an intended goal or approach of the initiative. If this is true for the authors, please elaborate and reference a greater breadth of literature (e.g. from the field of knowledge co-production/actionable science). If not, these references might be removed as they appear tangential to the central arguments/experiences of the authors and this work.

5) The reference to Witte 1992, as suggested by needs to be elaborated on. As currently incorporated, it is unclear how the authors see this theory as part of their approach or work.

6) The abstract still needs to be adapted to support the main arguments of the paper. Both reviewers suggested a reframing of the abstract, but no edits have been made. Given the significant changes to the manuscript, changes to the abstract are warranted.

7) Both reviewers suggested a new title. The framing around the 'post-factual' society, while more thorough in this version of the manuscript, still does not seem to be the central thesis of the paper. The last section of the paper "an adapted scientist" might be something the authors consider incorporating into the title.

8) Along those lines, the suggestions for the adapted scientist seem out of context at the end of the paper. These suggestions, and more explicit examples of how the Pole to Paris effort carried out these suggestions could be better integrated into the manuscript so it is clear how what was done in the initiative led to, or modeled, these suggestions. Further, the authors note in passing that the analysis of the social media analytics occurred well-after the experience, when some data from their social platforms were no longer available. Do the authors have any insights for others regarding data collection, audience segmentation, or documentation that they think would help 'the adapted scientist'? This is noted on page 8 and might be worthwhile incorporating more thoughtfully in their recommendations at the end of the manuscript.

9) As noted by the reviewers, a detailed proof-read is required to tighten up the manuscript.

While these suggestions warrant further work, I am confident this manuscript will provide an important published case study from a unique effort that provides evidence for the important role that climate scientists can play in more deeply, effectively engaging a broad set of audiences in climate change science and climate action.

---

## Author Response (AR2)

**Adapted climate scientists to a post-factual society: Reflections from the awareness campaign Pole to Paris**

Erlend M. Knudsen and Oria J. de Bolsée

**Set-up of response**

We thank the editors for their suggestion on a new title of our manuscript, as well as feedback on our response throughout the review process. We believe that the changes we have made in line with the comments by the reviewers have strengthened the manuscript overall.

In the following, we go through each comment by the editors (reproduced here in grey text for your reference) and explain our choices of changes in accordance with these. Where changes to the text in the manuscript are made, the relevant excerpt is reproduced from the .pdf manuscript to this .docx response in *italic text*, with changes written in *italic green text*.

**Response to the reviews by Heidi Roop and Sam Illingworth**

Renaming of the title
I still suggest that the title be changed to provide clarity on the core content of the manuscript, which is a discussion of the approach, methods and outcomes of the Pole to Paris initiative, along with some suggestions for other scientist-communicators based on your lessons learned.

As discussed by the editor, please can you provide a new title that clearly outlines the contents of the paper. "Communicating climate change in a "post-factual" society: Lessons learned from the Pole to Paris campaign" seems like an excellent suggestion, but please feel free to suggest something else that is equally as clear.

Based on your suggestions, we have changed the title from "Adapted climate scientists to a post-factual society: Reflections from the awareness campaign Pole to Paris" to "Communicating climate change in a "post-factual" society: Lessons learned from the Pole to Paris campaign."

**Changes by authors**

Based on the feedback from the research community that we have received since the manuscript re-submission, we have made some minor changes to the manuscript. We hope that the reviewers share our notion that these modifications are improving the manuscript without changing its content.

In the following, we highlight the changes to the second submitted version of the manuscript.

Reference to Paasche and Åkesson (2019)
After the previous submission, a highly relevant text for the messages in this manuscript was written by Paasche and Åkesson (2019). While this was an opinion piece, it was published in Eos, a source for trustworthy news and perspectives about the Earth and space sciences and their impact by the American Geophysical Union, and therefore provides more scientific support for some of the statements in our manuscript.

As a result, we have added references to Paasche and Åkesson (2019) in relevant excerpts in the 2nd and 3rd paragraphs in Sect. 3 and in the 2nd paragraph in Sect. 5. These now read:
*Since then, Donald Trump was elected as the country's 45th president and has repeatedly been questioning climate science, actively working against environmental legislation and funding of his predecessor and generally making the work of climate scientists more challenging (De Pryck and Gemenne, 2017; Alderman and Inwood, 2018; and references therein; Paasche and Åkesson, 2019, and references therein).*
*A post-factual political scene is not isolated to the U.S. alone; Brexit in the U.K. and the (re-)elections of Rodrigo Duterte in the Philippines, Andrzej Duda in Poland, Viktor Orbán in Hungary, Recep Tayyip Erdogan in Turkey and Jair Bolsonaro in Brazil are all examples of populistic solutions trumping science-based ones (Postel-Vinay, 2017; Paasche and Åkesson, 2019).*
*However, current training of becoming scientists does not fulfil society's current need for clear science communication and policy engagement (Leshner, 2007; Paasche and Åkesson, 2019).*

The new reference is:
● Paasche, Ø., and Åkesson, H.: Let's start teaching scientists how to withstand attacks on fact, Eos, 100, doi:10.1029/2019EO118499, 2019.

Data transparency
To allow for better data transparency in a scientific manner, we have published the data used in this manuscript in the open-access repository Zenodo. This publication, Knudsen and de Bolsée (2019), contains the same versions of Tables A1-A3 and Fig. A1 that was uploaded as supplementary material in the previous submission.

As a result, we have replaced the references to Tables A1-A3 and Fig. A1 in the supplementary material by references to this publication. The relevant excerpts in the 8th paragraph in Sect. 2 and the 1st and 7th paragraphs in Sect. 3 now read:
*A conservative estimation is that more than one million people in 45 countries were reached through conventional and social media, which included 252 media outlets (thereof 15*

*blog posts written by us; Knudsen and de Bolsée, 2019) and almost 500,000 and 250,000 reached per Facebook post and Twitter tweet, respectively.*

*Detailed data on these can be found in Knudsen and de Bolsée (2019).*

*The survey was set up through the online survey platform SurveyMonkey and asked the anonymous respondents a range of questions. These included whether respondents followed Pole to Paris online, whether they learned anything new as a result of Pole to Paris and whether they found Pole to Paris to be a source of inspiration (Knudsen and de Bolsée, 2019).*

Furthermore, we have added a new section between Sect. 5 and Author contributions named Data availability. It reads:

*The data used in this study is available in Knudsen and de Bolsée (2019).*

The new reference is:
- Knudsen, E. M., and de Bolsée, O. J.: Data used in "Communicating climate change in a 'post-factual' society: Lessons learned from the Pole to Paris campaign" (Version 2) [Data set], Zenodo, doi:10.5281/zenodo.2659211, 2019.

Acknowledgement of StormGeo

As the peer review process took place after EMK switched research positions from being at a university to being in a company, we find it important to also acknowledge the company he currently works for. StormGeo has been supporting his work through mentions in own channels and some allocated time to work on the responses.

*We would like to thank the whole Pole to Paris team and everyone we encountered along the way. We also wish to thank the two reviewers and editors for constructive suggestions that improved the manuscript. We gratefully acknowledge the funding by StormGeo and the German Research Foundation (Deutsche Forschungsgemeinschaft; DFG) for the Transregional Collaborative Research Center "ArctiC Amplification: Climate Relevant Atmospheric and SurfaCe Processes, and Feedback Mechanisms (AC)3" (TRR 172, project no. 268020496), which allowed EMK to write about our experiences for the benefit of science communicators globally.*

Video abstract

Similar to Data transparency above, we have published the video abstract for this manuscript in the open-access repository Zenodo. This publication, Knudsen (2019), contains the same message as in Video A1 that was uploaded as supplementary material in the previous submission and is therefore removed from the supplementary material.

The new reference is:

[revised manuscript text omitted]

**(a) *All organic vs. paid views**

346    56 130    16 703

| | >3 s | ≥30 s | 95 % |
| Paid | 42 % | 37 % | 26 % |
| Organic | 58 % | 63 % | 74 % |

**(b) *Average topical views**

897    4 868    1 437

| | >3 s | ≥30 s | 95 % |
| Journeys | 6 % | 6 % | 11 % |
| Climate change | 11 % | 7 % | 8 % |
| Climate action | 83 % | 87 % | 82 % |

**Figure 3: Percentages of total Facebook video views after three seconds (*>3 s*), at 30 seconds (or to the end, whichever came first; *≥30 s*), and at 95 % of the video length (including people that skipped to this point; *95 %*) for (a) organic (i.e., not paid; blue columns) and paid (red columns) views and (b) videos on climate action (blue columns), climate change (red columns) and the journeys themselves (green columns). Numbers above the columns in (a) and (b) represent total and average views, respectively.**